# Differently Prepared PbO$_2$/Graphitic Carbon Nitride Composites for Efficient Electrochemical Removal of Reactive Black 5 Dye

Aleksandar Marković [1], Slađana Savić [1], Andrej Kukuruzar [1], Zoltan Konya [2,3], Dragan Manojlović [1,4], Miloš Ognjanović [5] and Dalibor M. Stanković [1,5,*]

1    Faculty of Chemistry, University of Belgrade, Studentski trg 12-16, 11000 Belgrade, Serbia
2    Interdisciplinary Excellence Centre, Department of Applied and Environmental Chemistry, University of Szeged, Rerrich Béla tér 1, H-6720 Szeged, Hungary
3    MTA-SZTE Reaction Kinetics and Surface Chemistry Research Group, Rerrich Béla tér 1, H-6720 Szeged, Hungary
4    South Ural State University, Lenin Prospekt 76, 454080 Chelyabinsk, Russia
5    Department of Theoretical Physics and Condensed Matter Physics, VINČA Institute of Nuclear Sciences, National Institute of the Republic of Serbia, University of Belgrade, Mike Petrovića Alasa 12-14, 11000 Belgrade, Serbia
*    Correspondence: dalibors@chem.bg.ac.rs

**Abstract:** In this paper, electrochemical degradation of Reactive Black 5 (RB5) textile azo dye was examined in regard to different synthesis procedures for making PbO$_2$–graphitic carbon nitride (g-C$_3$N$_4$) electrode. The reaction of Pb(OH)$_3^-$ with ClO$^-$ in the presence of different surfactants, i.e., cetyltrimethylammonium bromide (CTAB) and tetrabutylammonium phosphate (TBAP), under conventional conditions, resulted in the formation of PbO$_2$ with varying morphology. The obtained materials were combined with g-C$_3$N$_4$ for the preparation of the final composite materials, which were then characterized morphologically and electrochemically. After optimizing the degradation method, it was shown that an anode comprising a steel electrode coated with the composite of PbO$_2$ synthesized using CTAB as template and g-C$_3$N$_4$, and using 0.15 M Na$_2$SO$_4$ as the supporting electrolyte, gave the best performance for RB5 dye removal from a 35 mg/L solution. The treatment duration was 60 min, applying a current of 0.17 A (electrode surface 4 cm$^2$, current density of 42.5 mA/cm$^2$), while the initial pH of the testing solution was 2. The reusability and longevity of the electrode surface (which showed no significant change in activity throughout the study) may suggest that this approach is a promising candidate for wastewater treatment and pollutant removal.

**Keywords:** reactive azo dye; surfactant-assisted synthesis; electrode morphology; advanced oxidation processes; lead dioxide; energy efficiency





## 1. Introduction

Reactive Black 5 (RB5) dye, or Remazol Black B (Scheme 1), belongs to the vinyl sulphone type of azo dyes. Azo dyes, aromatic compounds with one or more –N=N– groups, constitute the largest class of synthetic dyes in commercial application [1,2]. RB5 is most commonly used to dye cotton and other cellulosic fibers, wool, and nylon [3,4]. This type of dye is highly soluble in water and has reactive groups which can form covalent bonds with fibers [4].

**Scheme 1.** Chemical structure of Reactive Black 5 azo dye.

Industries such as textile, leather, paper, pulp, printing, and dyeing are major consumers of synthetic dyes. Such industries produce large quantities of colored wastewater which cause significant adverse effects on the environment. The release of colored wastewaters is not only aesthetically unpleasant, but also obstructs penetration of light, which affects biological processes. It can also potentially generate toxicity for aquatic organisms and finally, via food chain or exposure, in humans [5–9].

Different methods for degradation and removal of RB5 can be found in scientific literature, such as biodegradation and biocatalysis [1,10,11], photocatalysis [12–15], electrocoagulation [2], Fenton or Fenton-like reaction [6,16], adsorption [9], or by activation of peroxymonosulfate (PMS) [17,18].

Each of these methods has its positive and negative sides. For instance, electrocoagulation, membrane separation processes, adsorption, and precipitation only change the phase of pollutants. Photo- and chemical oxidation require additional chemicals and oxidation agents that are considered highly toxic and can produce additional hazardous waste. Photochemical oxidation is one of the most commonly used methods for the degradation of waterborne pollutants [19]. $TiO_2$-based materials are most widely used because of their unprecedented photocatalytic activity, but other metallic and nonmetallic catalysts are giving promising results [20–22]. Biodegradation can yield very good results but it can also be less effective than other methods, because dyes can be toxic for bacteria and can thus inhibit their activity.

Electrochemical oxidation processes present an effective and logical choice for the development of green, time-effective, and, at the same time, potent methods for the removal of various pollutants, including reactive textile dyes [23–25]. Electrocatalytic processes can be considered green primarily due to mild and environmentally acceptable working conditions (ambient pressure and temperature, work in aqueous media, no additional toxic chemicals). These methods may also be coupled with renewable energy sources [26], they do not require systems for temperature or pressure control nor expensive gases and give the possibility of avoiding expensive catalysts based on precious metals, so their potential for large-scale application is enormous.

Therefore, electrochemical methods, through the rational design of the electrocatalytic setup, enable simple and practical systems for water purification and pollutant removal. Our research group proposed several approaches for the efficient removal of organic pollutants using electrooxidation methods, such as for the removal of Reactive Blue 52 [27,28], triketone herbicides [29], or ibuprofen [30].

For electrochemical oxidative processes, anodes with high oxygen evolution potentials (non-active anodes) achieve complete mineralization of organics and are thus favored for wastewater remediation. The most commonly used non-active anode is the boron-doped diamond electrode, but cheaper alternatives include various metal oxides such as $TiO_2$, $SnO_2$, and $PbO_2$ [31]. Metal oxides exhibit exceptional properties valuable to electrochemical processes, while featuring reduced costs, availability and environmental compatibility, and have thus found diverse applications. The properties of metal oxides

can be fine-tuned by controlling their morphology, particle size, and crystallinity, which can be achieved using template synthesis.

Lead dioxide is an inexpensive and widely available material, readily incorporated in composite materials with outstanding properties [32]. Lead oxide-based anodes gave promising performances for electrochemical degradation of organic pollutants [33], as they can possess unique catalytically active surfaces [34], good stability, and a long electrode life [35].

Graphitic carbon nitride (g-$C_3N_4$) has found numerous applications owing to its properties, such as high surface area, low cost, and biocompatibility. The use of g-$C_3N_4$ in composites proved to enhance its photo- and electrocatalytic properties. Good results for the photocatalytic degradation of organic pollutants were achieved when using nanocomposites made from metal oxides and g-$C_3N_4$ [36–39]. On the other hand, reports of g-$C_3N_4$ being used for water remediation in a purely electrochemical setting are scarce.

The main objective of this paper was to investigate the effects of different surfactant templates in the synthesis of $PbO_2$, to prepare of composite $PbO_2$/g-$C_3N_4$, and to optimize working conditions for electrocatalytic degradation of RB5 using a coated stainless-steel anode. The conditions in question include the solution pH value, current density (applied voltage) during the reaction, and concentrations of both the supporting electrolyte and RB5. The electrocatalytic properties of $PbO_2$ synthesized with cetyltrimethylammonium bromide (CTAB) as template (hereafter abbreviated to $PbO_2$-CTAB) vs. $PbO_2$ synthesized with tetrabutylammonium phosphate (TBAP) as template (hereafter abbreviated to $PbO_2$-TBAP), as well as their composites with graphitic carbon nitride (hereafter abbreviated to $PbO_2$-CTAB/g-$C_3N_4$ and $PbO_2$-TBAP/g-$C_3N_4$) were investigated. Moreover, the morphological and electrocatalytic properties of these materials were studied. Finally, the best combination of conditions was used for the real-world water sample treatment, artificially polluted with RB5.

## 2. Results and Discussions

The present research was designed to determine whether the stainless steel (SS) electrodes coated with composites made from $PbO_2$ and g-$C_3N_4$ could be applied for RB5 electrochemical decoloration.

The electrode materials were based on $PbO_2$ nanoparticles, synthesized with CTAB and TBAP surfactants as templates, later combined with g-$C_3N_4$ to obtain composites tested in the removal of the selected textile dye. To optimize the electrochemical procedure, the initial pH value, the supporting electrolyte concentration, the current density, and the RB5 concentration were varied. Finally, the composites were compared in terms of efficiency of RB5 removal, under previously optimized conditions. Additionally, the stability, longevity, and energy consumption were also reported and set side by side with the literature data.

### 2.1. Morphological and Electrochemical Properties of Materials

Microstructure characterization and phase analysis were performed by powder X-ray diffraction (PXRD) (Figure 1A). The $PbO_2$-CTAB/g-$C_3N_4$ crystallizes in group P42/mnm, which can be ascribed to the β-$PbO_2$ (JCPDS #89-2805) crystal phase. The main diffraction peaks appeared at 2θ = 25.4°, 32.0°, 36.1°, and 49.1° which can be associated with the (110), (101), (200), and (211) standard diffraction peaks of tetragonal β-$PbO_2$ [40]. The remaining small intensity reflections can be assigned to α-$PbO_2$, as a small part of the sample crystallizes in the Pbcn space group [41]. On the other hand, $PbO_2$-TBAP/g-$C_3N_4$ (red line) crystallizes both in alpha and beta lead oxides. From the PXRD patterns, it is also clear that the intensity of diffraction peaks is weaker than for $PbO_2$-CTAB, indicating lower crystallinity of $PbO_2$-TBAP.

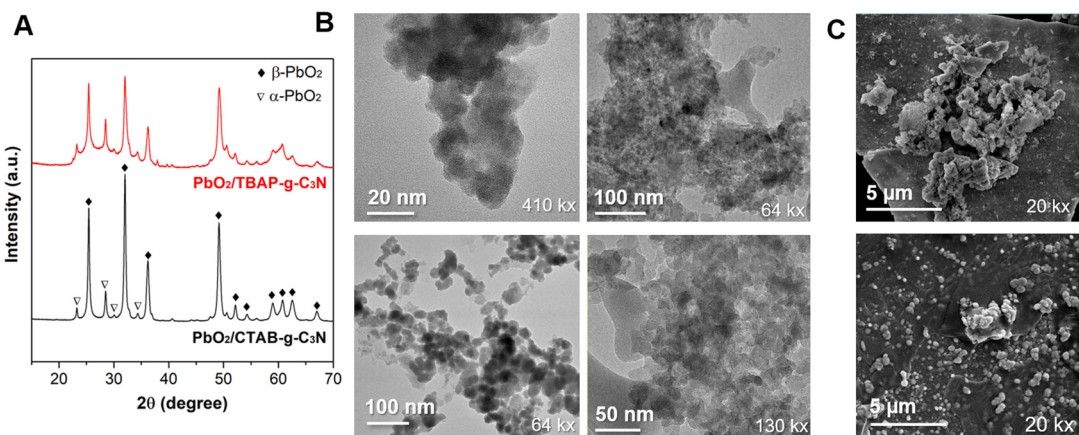

**Figure 1.** Morphological and microstructural analysis. (**A**) PXRD profiles of PbO$_2$-CTAB/g-C$_3$N$_4$ and PbO$_2$-TBAP/g-C$_3$N$_4$, (**B**) TEM micrographs of PbO$_2$-CTAB (upper left), PbO$_2$-CTAB/g-C$_3$N$_4$ (upper right), PbO$_2$-TBAP (lower left), PbO$_2$-TBAP/g-C$_3$N$_4$ (lower right) and (**C**) FE-SEM micrographs of PbO$_2$-CTAB/g-C$_3$N$_4$ (up) and PbO$_2$-TBAP/g-C$_3$N$_4$ (down).

The morphology and size analysis of the prepared catalysts was performed using electron microscopy. The transmission electron microscopy (TEM) images of the composite materials are displayed in Figure 1B. As can be seen, particles of both PbO$_2$-CTAB and PbO$_2$-TBAP are of elongated spherical to irregular shape, partially agglomerated, with not-so-distinctive boundaries between particles. The average particle size of PbO$_2$-CTAB is about 10 nm, while the PbO$_2$-TBAP particles are twice as large (~20 nm). The micrographs of composites with g-C$_3$N$_4$ reveal that smaller particles are interconnected with large sheets of graphitic carbon nitride that enable a larger specific surface. For further analysis of composite materials, field emission scanning electron microscopy (FE-SEM) measurements of PbO$_2$-CTAB/g-C$_3$N$_4$ and PbO$_2$-TBAP/g-C$_3$N$_4$ are shown in Figure 1C. In these micrographs, it can be seen that the small PbO$_2$ nanoparticles are scattered all over the sheet of g-C$_3$N$_4$. The oxide particles in the PbO$_2$-TBAP/g-C$_3$N$_4$ sample are better dispersed on the carbon nitride surface, while the PbO$_2$-CTAB/g-C$_3$N$_4$ sample particles are more agglomerated on top of the g-C$_3$N$_4$ sheet.

To confirm potential applicability of the materials for the removal of organic pollutants from wastewaters using electrochemical advanced oxidation processes, electrocatalytic characterization of the materials was conducted. Electrochemical properties of the materials were scrutinized by employing cyclic voltammetry (CV) and electrochemical impedance spectroscopy (EIS) in Fe$^{2+/3+}$ solution in 0.1 M KCl. Results are depicted in Figure 2. As can be seen, PbO$_2$-CTAB provides well defined and oval shaped redox peaks of the Fe$^{2+/3+}$ couple. Using the composite material PbO$_2$-CTAB/g-C$_3$N$_4$ resulted in a significant increase in the redox couple current, with negligible or no changes in peak potentials. This behavior indicates that the prepared composite significantly improves electrocatalytic properties of the electrode regarding diffusion and mass transfer. In the case of PbO$_2$-TBAP, it is noticeable that additional peaks are present, which can be attributed to the metal oxide modifier. In the case of the PbO$_2$-TBAP/g-C$_3$N$_4$ composite, its successful formation can be confirmed by the absence of additional peaks. However, a significant difference in peak potentials can be assigned to the lower mass transfer ability. Similar to these results, by employing impedance spectroscopy, we can conclude that the involvement of graphitic carbon nitride in the composite structure of PbO$_2$-CTAB/g-C$_3$N$_4$ results in increased diffusion capabilities at the electrode/solution interface, based on the linear part of the spectra, while the decrease in the $R_{ct}$ value in the semicircle region is negligible.

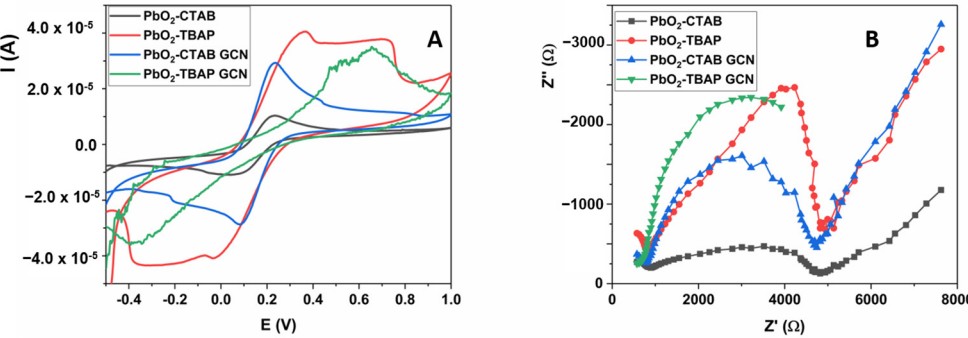

**Figure 2.** (**A**) CV voltammograms of modified electrodes in $Fe^{2+/3+}$ redox couple at the scan rate of 50 mV/s; (**B**) EIS spectra of modified electrodes in $Fe^{2+/3+}$ redox couple. Working potential 0.1 V.

### 2.2. Application of Composites for the Electrochemical Removal of RB5

The initial phase of this study was selecting the optimal operating parameters to validate the applicability of the $PbO_2/g$-$C_3N_4$ composite on SS electrode in a new electrochemical degradation method for real water treatment, to ultimately exploit all its advantages and possibilities. In this regard, we optimized several important parameters: initial pH, supporting electrolyte ($Na_2SO_4$) concentration, current density-applied voltage, and the initial concentration of the selected pollutant.

2.2.1. pH Optimization

Starting pH of the supporting solution represents one of the most important factors for the performance of electrochemical processes. In order to determine the optimal initial pH for electrochemical degradation, RB5 was dissolved in 0.1 M $Na_2SO_4$ to make a 70 mg/L solution with native pH 5.63 (measured using pH meter). The $\lambda_{max}$ of this solution obtained by the spectrometer was 598 nm. Three initial pH values were tested—2, 4, and 6. The native pH of the stock solution was adjusted to these values using 0.1 M NaOH and 0.1 M $H_2SO_4$. The electrochemical reaction was allowed to continue almost to the point when the absorbance on the UV/Vis spectrum reached a plateau. For the reaction, one stainless steel (SS) electrode (4 $cm^2$ of surface area) modified with $PbO_2$-CTAB/g-$C_3N_4$ was used as the anode while an unmodified SS electrode was used as the cathode. The experiments were performed under constant a voltage of 5 V and the results are shown in Figure 3.

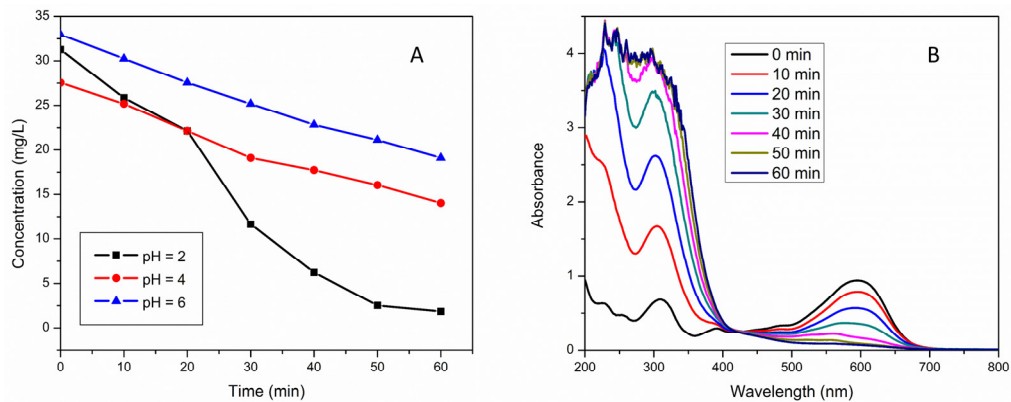

**Figure 3.** (**A**) RB5 removal under pH 2, 4, and 6; (**B**) UV–Vis spectra of RB5 solution in the range from 200 to 800 nm during 60 min of treatment (pH 2, 0.1 M $Na_2SO_4$, 5V) at different time intervals.

As can be seen in Figure 3A, the best degradation rate was observed at initial pH 2. After 60 min of treatment using this starting pH, 98% of the color was removed. This can be attributed to the cleavage of the –N=N– bonds and aromatic rings, resulting in the decrease of optical density of the dye solution (Figure 3B).

Moreover, in the acidic medium, $OH^\bullet$ radicals are produced by the anodic discharge in water in the indirect electrochemical oxidation of organic dyes at the anode. These $OH^\bullet$

radicals adsorb onto the anode surface and oxidize the organic material. At higher pH, it could be expected that a larger amount of hydroxyl radicals would be generated, which would result in greater efficiency of the system. However, it was determined that the degree of ionization of the cationic dye is highly dependent on the initial pH value of the solution, which most often leads to the formation of a precipitated photochromic compound and a decrease in resistance on mass transfer at higher pH. Therefore, an acidic environment leads to better dissolution and degree of ionization. In a basic environment, an elevated consumption of electrolytes occurs dominantly, which directly affects the conductivity of the solution. Lower pH values were not tested as recent studies showed that low pH values lead to a higher dissolution rate of lead dioxide [42] and that these values are not appropriate for work with lead dioxide films, as they can cause the formation of lead sulfate [43].

The occurrence of new peaks In the spectrum confirms the formation of smaller molecules. The increase in absorbance for these peaks was followed by the decrease of the main RB5 peak, suggesting a relationship between these processes and successful decomposition of RB5. Thus, based on the conducted study, all further experiments were done using starting pH 2 for the dye solutions.

### 2.2.1. Optimization of Supporting Electrolyte Concentration

As mentioned earlier, $Na_2SO_4$ was used as the supporting electrolyte. This choice was made based on the fact that this salt is of low cost, is often found in textile industry wastewater in high concentrations, has excellent conductive properties, and, most importantly, is inert under oxidative conditions, so that the degradation efficiency can entirely be attributed to the developed method. The function of the supporting electrolyte is to improve the electrical conductivity and current transfer and presents one of the most important factors in electrochemical processes. In this study, we investigated the effect of different concentrations of supporting electrolytes, testing the following concentrations: 0.01 M, 0.05 M, 0.1 M, 0.15 M, and 0.2 M (Figure 4). Other applied conditions were pH 2, constant voltage (5 V), and RB5 concentration of 35 mg/L.

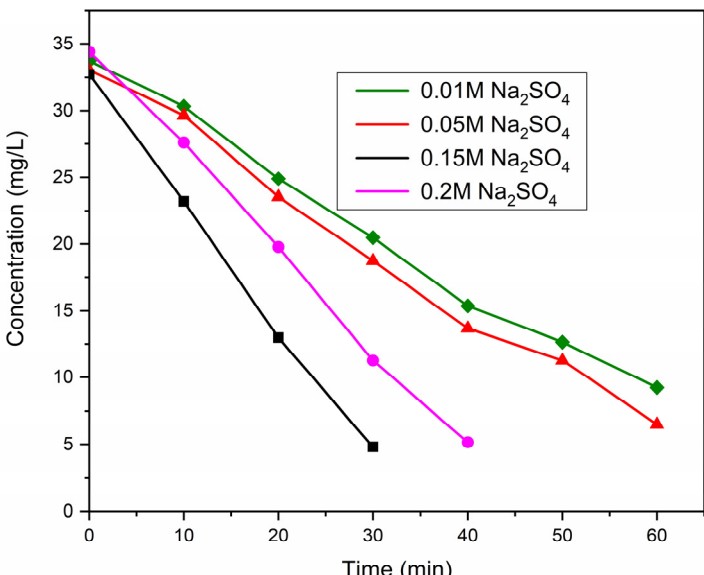

**Figure 4.** Degradation of RB5 dye under pH 2 and different concentrations of $Na_2SO_4$.

As expected, the increase in electrolyte concentration was followed by a more rapid removal rate. Up to 0.15 M of sodium sulfate, these changes were more noticeable, while when the concentration was increased to 0.2 M, the removal rate slightly diminished. Belal and coworkers studied the effect of supporting electrolytes and their concentrations on dye removal efficiency and found a similar trend [44]. The decrease in removal rate can

be connected with the generation of fewer oxidants at higher electrolyte concentrations, due to increased production of $S_2O_8^{2-}$ from sulfate ions. It should be noted that in all the probes using the selected concentrations, the decolorization of RB5 was carried out up to nearly 100%. Therefore, to further extend the optimization of the experimental conditions we selected the supporting electrolyte concentration of 0.15 M, at the previously optimized initial pH of 2.

### 2.2.2. Optimization of Current Density-Applied Voltage

In previous optimization experiments, the voltage was held constant (at 5 V) and the current was constantly changing, depending on the duration of the experiment. In this study, probes were made under the opposite conditions—the current was held constant while the voltage was allowed to vary, but it was still monitored. The currents used for the probes were 0.08 A, 0.10 A, 0.13 A, 0.17 A, and 0.20 A (Figure 5). These currents were applied on an electrode with a working area of 4 $cm^2$, therefore the calculated current densities were as follows: 20 mA/$cm^2$, 25 mA/$cm^2$, 32.5 mA/$cm^2$, 42.5 mA/$cm^2$, and 50 mA/$cm^2$. The rest of the conditions were pH 2, 0.15 M $Na_2SO_4$, and 35 mg/L of RB5.

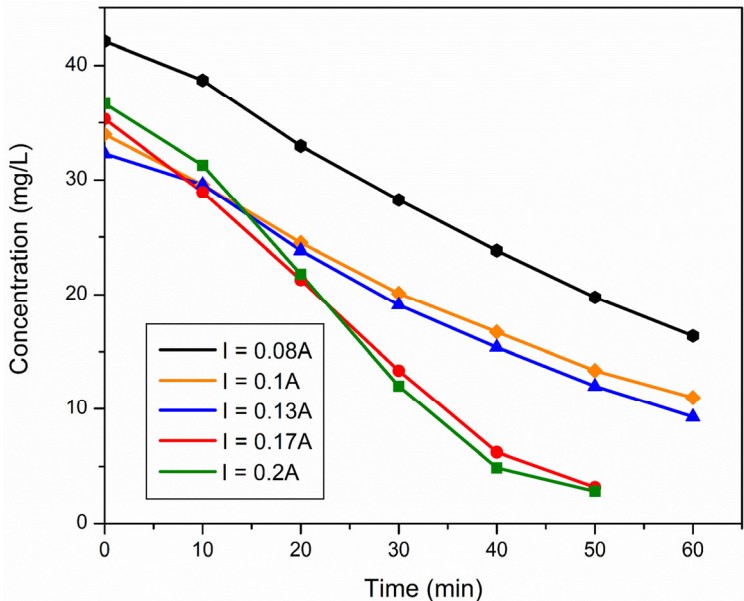

**Figure 5.** Degradation of RB5 dye under pH 2, 0.15 M of $Na_2SO_4$, and different applied currents.

During the experiments, the voltage was monitored because this information can be important for the possible construction of pilot reactors and further improvement of the method. Results pertaining to the relationship between current densities and the observed voltages are summarized in Table 1.

**Table 1.** Corresponding values of voltage when current was held constant.

| I (A) | U (V) |
|---|---|
| 0.08 | ~3.63 |
| 0.10 | ~3.65 |
| 0.13 | ~4.2 |
| 0.17 | ~4.8–4.9 |
| 0.20 | ~5.1–5.3 |

An applied voltage ranging from 3.6 V to 5.3 V, with an output current that is less than or equal to 200 mA, meets the conditions required for possible future technology transfer. Additionally, the low values of applied voltage and current density do not cause significant temperature changes of the test solution. Namely, after the treatment, the increase in

solution temperature was lower than 3 °C for all optimization experiments. The results were as expected, since increasing the current density accelerated the decolorization of RB5. This occurs up to the current density of 42.5 mA/cm$^2$ (applied current 0.17 A $\rightarrow$ potential 4.8–4.9 V). Increasing the current to 0.2 A does not cause a significant improvement in efficiency. This can be explained by side reactions, where sulfate ions migrate to the electrode surface, occupy active sites and reduce the effective surface area. This ultimately results in surface inactivation and reduced contaminant removal capacity.

### 2.2.3. Optimization of RB5 Dye Concentration

It is known that industrial wastewater usually contains different concentrations of textile dyes, depending on its nature and reactivity. Moreover, it is known that RB5 is distinguishable in water at 1 mg/L concentration with the naked eye while its concentrations in the effluents of industries are usually in the range of 10–100 mg/L [45]. From this, it can be concluded that evaluating the initial concentration of RB5 in the removal process is a very important aspect. Based on that, we investigated the performance of our method in the removal of different amounts of RB5 in the working solution. Concentrations of RB5 that were examined were 20 mg/L, 35 mg/L, 70 mg/L, and 100 mg/L. The rest of the conditions were as previously optimized: pH 2, 0.15 M Na$_2$SO$_4$, and applied current 0.17 A. The results of these measurements are presented in Figure 6. It is noticeable that increasing the concentration of RB5 causes a decrease in efficiency for this method. When comparing the performance in lower concentrations, it can be concluded that for an increase in the amount of dye from 20 to 35 mg/L, treatment time only had to be slightly prolonged (in order to achieve the same degradation efficiency), while further increasing the dye concentration required extended treatment times. Based on the performed study, the dye concentration of 35 mg/L could be considered as optimal for further experiments.

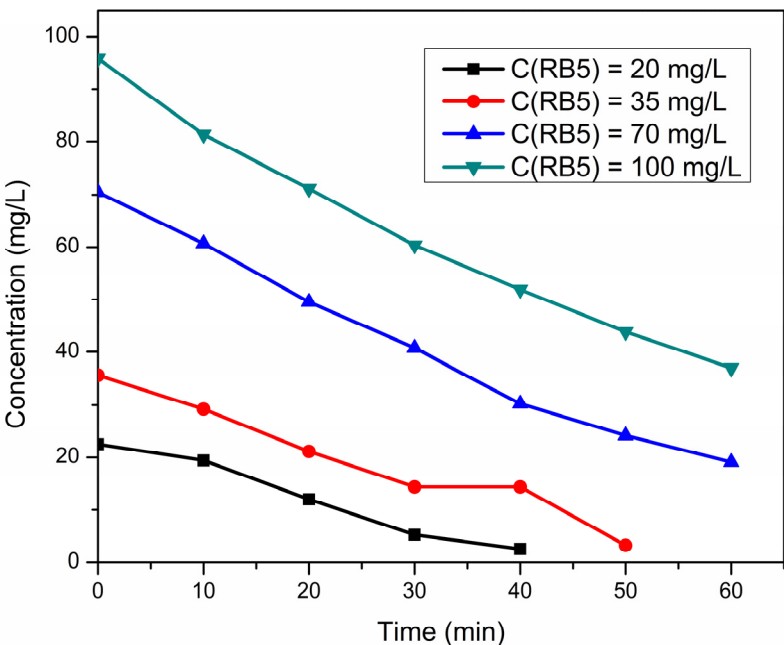

**Figure 6.** Degradation of RB5 dye under pH 2, 0.15 M Na$_2$SO$_4$, I = 0.17 A, and different initial concentrations of RB5 ranging from 20 mg/L to 100 mg/L.

### 2.3. Performance Comparison of the Synthesized Materials and Composites

In order to determine the electrocatalytic properties of the prepared materials and anodes, we tested their performances in a 35 mg/L solution of RB5 under fully optimized conditions: initial pH 2, current density 42.5 mA/cm$^2$, and supporting electrolyte concentration of 0.15 M. The results are summarized in Figure 7. As can be seen, the introduction of graphitic carbon nitride in the materials and formation of composites strongly influenced

the results of the developed method. This can be assigned to the previously reported medium bandgap value for g-C$_3$N$_4$ and its characteristic catalytic performances [46–52]. However, it is obvious that the morphology of lead oxide is strongly correlated with the catalytic properties of g-C$_3$N$_4$ [53–55]. In the studied group of materials, the nanocomposite prepared from lead dioxide synthesized with assistance of linear surfactant cetyltrimethy-lammonium bromide as template and graphitic carbon nitride showed superior properties for electrocatalytic application, since using this electrode material led to almost 90% removal of RB5 in an hour.

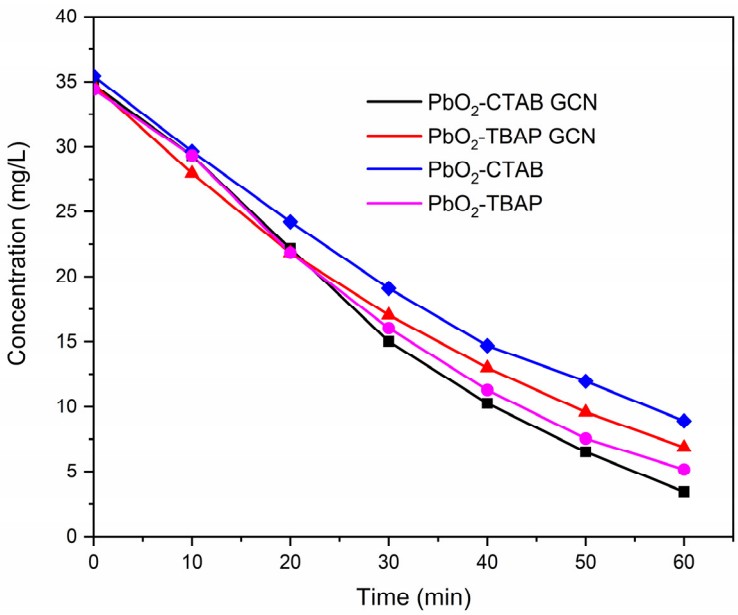

**Figure 7.** Degradation of RB5 dye under pH 2, Na$_2$SO$_4$ concentration of 0.15 M, I = 0.17 A, RB5 concentration of 35 mg/L, using different anodes.

Table 2 provides an overview of different electrochemical systems employed to remove RB5, with focus on results with Na$_2$SO$_4$ as supporting electrolyte. All selected studies reported faster RB5 removal when using NaCl or KBr [4,56–58], but the use of these non-inert salts is discouraged in electrochemistry, because they lead to the formation of chlorinated and brominated degradation products, which are often more toxic compared to the parent compound [57,59,60].

**Table 2.** The RB5 degradation efficiency comparison with the literature data.

| Material | Supporting Electrolyte | % of RB5 Removal and Time | Energy Consumption | Ref |
|---|---|---|---|---|
| RuO$_2$/IrO$_2$/TiO$_2$@DSA$^{®}$ | 0.008 M NaCl | ~100% in 15 min | Not reported | [4] |
| Ti/CoO$_x$–RuO$_2$–SnO$_2$–Sb$_2$O$_5$ | 0.07 M Na$_2$SO$_4$ | ~40% in 2 h | 34.5 kWh/kg [a] | [56] |
| Ti/SnO$_2$-Sb$_2$O$_5$-IrO$_2$ | 0.1 M Na$_2$SO$_4$ | ~60% in 2 h | Not reported | [58] |
| Graphite | 0.1 M Na$_2$SO$_4$ | ~100% in 3 h | Not reported | [57] |
| PbO$_2$-CTAB/g-C$_3$N$_4$ on SS | 0.15 M Na$_2$SO$_4$ | ~90% in 1 h | 0.4374 kWh/g | This study |

[a] calculated as general current efficiency, based on COD removal [61].

When excluding the results with NaCl, our study gave the fastest (under 1 h) and the highest RB5 removal (nearly 90%). The system with Ti/CoO$_x$–RuO$_2$–SnO$_2$–Sb$_2$O$_5$ removed only 40% of the selected dye, during 2 h of treatment in the presence of Na$_2$SO$_4$, but increased to over 95% when NaCl was used [56]. The next study with Ti/SnO$_2$-Sb$_2$O$_5$-IrO$_2$ was slightly improved in the same time interval without the addition of NaCl [58]. Graphite electrode led to complete RB5 degradation and quite efficient COD removal, but this experimental setup lasted for three hours [57].

### 2.4. Specific Energy Consumption

The specific energy consumption of an advanced oxidation process is an important indicator of economic efficiency, taking into account that a major part of operating costs for the proposed procedure derives from electric energy consumption. For this reason, the average energy consumption for a 60 min treatment was calculated when using each of the four prepared electrodes. This study was performed under all the previously optimized parameters. The results are listed in Table 3. As can be seen, $PbO_2$-CTAB in synergy with graphitic carbon nitride showed the lowest energy consumption during treatment of RB5 and can potentially be used for designing electrochemical systems for the removal of organic pollutants.

**Table 3.** Specific energy consumption (SEC) during RB5 treatment, a comparison of different electrode combinations.

| T (min) | 0 | 10 | 20 | 30 | 40 | 50 | 60 | |
|---|---|---|---|---|---|---|---|---|
| electrode | | RB5 degradation rate (%) | | | | | | SEC (kWh/g) |
| I | 0.00 | 16.45 | 37.43 | 57.69 | 70.19 | 81.01 | 88.53 | **0.4374** |
| II | 0.00 | 15.81 | 31.31 | 45.49 | 58.15 | 67.64 | 74.89 | **0.5173** |
| III | 0.00 | 19.94 | 37.52 | 50.93 | 62.79 | 72.14 | 80.24 | **0.4895** |
| IV | 0.00 | 14.83 | 36.57 | 53.38 | 67.21 | 78.09 | 85.01 | **0.4698** |

**I**—$PbO_2$-CTAB/g-$C_3N_4$ on SS; **II**—$PbO_2$-CTAB on SS; **III**—$PbO_2$-TBAP/g-$C_3N_4$ on SS; **IV**—$PbO_2$-TBAP on SS.

### 2.5. Stability and Longevity of the Method

To scrutinize stability and longevity of the method we monitored electrode efficiency during 7 days of testing. During that period, we performed 5 to 7 experiments per day. Each experiment lasted for 60 min (Figure 8). The relative standard deviation between absorbances was lower than 7% indicating that the proposed electrode possesses excellent stability and durability during this time period.

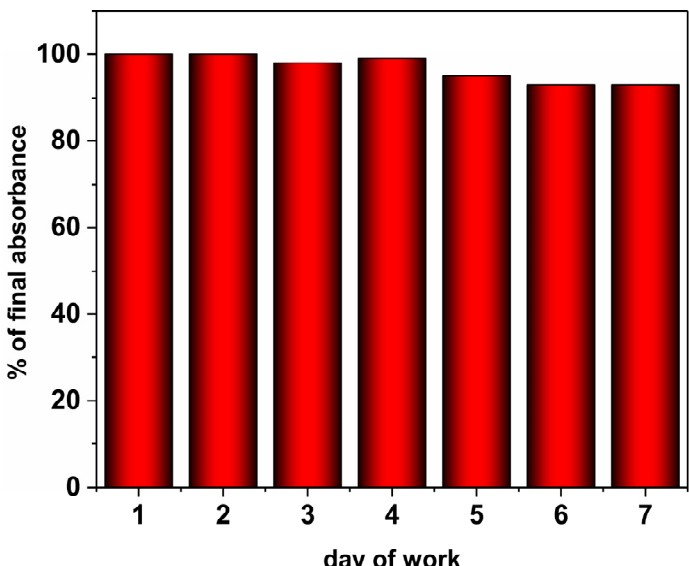

**Figure 8.** Changes in the final absorbance during seven days of work with the same electrode.

## 3. Materials and Methods

### 3.1. Chemicals

Reactive Black 5 dye (M-991.82, CAS number 17095-24-8, chemical purity grade, dye content ~50%) was purchased from Sigma-Aldrich (St. Louis, MO, USA). Cetyltrimethy-lammonium bromide (CTAB), tetrabutylammonium phosphate (TBAP), sodium hydroxide,

hydrochloric acid, sulfuric acid, ethanol (96%), acetone, urea, sodium hypochlorite, lead nitrate, dimethylformamide, and sodium sulfate were p.a. grade and purchased from Sigma-Aldrich. All chemicals were used as supplied without any purification. All the solutions were prepared in double distilled water. Working solutions were prepared before every experiment and used accordingly. Calibration solution for UV/Vis spectrometry was prepared using the supporting electrolyte.

### 3.2. Material Synthesis

Graphitic carbon nitride (g-$C_3N_4$) was synthesized from urea. Briefly, 25 g of urea was heated in air in a ceramic crucible with a lid, starting from room temperature up to 550 °C, and this temperature was maintained for 4 h. $PbO_2$ was synthesized using the procedure proposed by Cao et al. [62]. In summary, CTAB or TBAP were added to a 0.015 M solution of $Pb(OH)_3^-$. After the resulting solution was stirred for 30 min at 50 °C, which ensured the complete dissolution of the surfactants, 1 mL of 1.5 M (10%) aqueous NaClO solution was added under constant stirring. This gave a homogeneous solution. After heating the solution at 85 °C for 3 h, the resulting black-brown precipitate was collected, washed several times with absolute ethanol and distilled water, centrifuged, and dried under vacuum at room temperature for 5 h [62]. The materials thus obtained were labeled $PbO_2$-CTAB and $PbO_2$-TBAP. The composite materials were prepared using the above-mentioned metal oxide materials and graphitic carbon nitride by ultrasonification—5 mg of $PbO_2$-CTAB or $PbO_2$-TBAP and 50 mg of g-$C_3N_4$ were suspended in 1 mL of dimethylformamide and ultrasonicated for 90 min. The prepared materials were labeled $PbO_2$-CTAB/g-$C_3N_4$ and $PbO_2$-TBAP/g-$C_3N_4$.

### 3.3. Electrode Preparation

To clean up and roughen the surface of the (2 cm × 5 cm) raw stainless-steel mesh, it was first washed ultrasonically in 1 M $H_2SO_4$ for 60 min and then in acetone for 40 min, fully rinsed with distilled water, and dried at 80 °C in air. A measurement of 20 μL of prepared nanomaterial suspensions (5 mg of $PbO_2$/CTAB or $PbO_2$/TBAP and 50 mg of g-$C_3N_4$ in 1 mL of dimethylformamide) was added to the electrode surface to cover its entire surface area. An electrode prepared in such a manner was allowed to dry overnight and the same procedure was applied on the opposite side of the electrode. The coated electrodes were used as anodes while the uncoated electrodes served as cathodes.

### 3.4. Electrochemical and Morphological Characterization

Microstructural and morphological characterization was performed using transition electron microscopy (TEM), scanning electron microscopy (SEM), and powder X-ray diffraction (PXRD). PXRD measurements were performed on Rigaku's high-resolution SmartLab® diffractometer equipped with a CuKα radiation source, an accelerating voltage of 40 kV, and a current of 30 mA. The diffraction patterns were recorded in the 10–70° 2θ range with a measurement speed of 1°/min and a step of 0.05°. The morphology of prepared $PbO_2$-CTAB/g-$C_3N_4$ and $PbO_2$-TBAP/g-$C_3N_4$ were examined by high-resolution transmission electron microscopy (HR-TEM, FEI Technai G2, Hillsboro, Thermo Fisher Scientific, Waltham, MA, USA) applying an accelerating voltage of 200 kV. The samples were prepared by doping diluted aqueous suspensions into copper grids (300 mesh carbon, Ted Pella Inc., Redding, CA, USA) and left to dry at RT. The composite materials were also analyzed with scanning electron microscopy (Hitachi S-4700) operating at 10 kV acceleration voltage, equipped with the X-ray spectroscopy (EDX) accessory (Röntec QX2 spectrometer). A few nm of gold was condensed on the sample surface to prevent them from becoming charged.

Electrochemical properties of the materials were examined using cyclic voltammetry (CV) and electrochemical impedance spectroscopy (EIS) measurements (done on CHInstruments model CHI760b, Austin, TX, USA) in the solution of 5 mM $Fe^{2+/3+}$ redox couple in 0.1 M KCl. Measurements were done in a three-electrode system, where material properties were tested using carbon paste electrodes (CPE) modified with 10% of the synthesized

nanomaterials and composites. As a reference electrode, a Ag/AgCl (3M KCl) electrode was used while a platinum wire was employed as an auxiliary electrode. EIS measurements were done using the same assembly in the range from 0.01 to $10^5$ Hz at the potential of 0.1 V.

### 3.5. Experimental Setup

Solutions of RB5 with $Na_2SO_4$ were made in concentrations ranging from 20 mg/L up to 100 mg/L. The blank probe for UV–Vis was made with 0.1 M $Na_2SO_4$. For pH adjustments, 0.1 M NaOH and 0.1 M $H_2SO_4$ were used.

The optimization of RB5 degradation experiments was held in a vessel mounted on a magnetic stirrer, with a constant starting volume of the treated solutions (60 cm$^3$ of RB5 dissolved in $Na_2SO_4$ solution). The electrodes were immersed in the solution and connected to the voltage source PeakTech 1525. For RB5 concentration monitoring, samples were withdrawn in 10-min intervals, up to 90 min of electrochemical oxidation, and analyzed by Evolution™ One/One Plus UV–Vis Spectrophotometer, Waltham, MA, USA.

The first parameter to be optimized was starting pH value (adjusted using $H_2SO_4$ and NaOH), followed by the optimization concentration of the supporting electrolyte, with a fixed pH value. When these two parameters were set right, in terms of the fastest RB5 degradation, the current density was varied to improve the degradation rate. The last criterion was the RB5 concentration and with the optimized previously stated values, the synthesized electrode materials were compared to distinguish the most excellent conditions for RB5 degradation. Data were processed using MS Excel 2016 and OriginPro 8.

### 3.6. Specific Energy Consumption

To deduce the most efficient conditions for degradation of RB5, the specific energy consumption (SEC; kWh/g) was calculated:

$$\text{SEC}_{RB5}\left(\frac{kWh}{g}\right) = \frac{I\ (A) \times E_{cell}(V) \times T\ (h)}{V\left(dm^3\right) \times \Delta C_{RB5}\left(\frac{g}{dm^3}\right) \times 1000}$$

where I (A) was the applied electrical current; $E_{cell}$ (V) the cell voltage, T (h) the treatment time, V (dm$^3$) the treated solution volume, $\Delta C_{RB5}$ (g/dm$^3$) the difference between the starting and final RB5 concentration, while a conversion factor of 1000 was needed to convert from Wh/g to kWh/g. This equation was adopted from elsewhere [61].

### 4. Conclusions

In this paper, an electrochemical degradation/removal method for Reactive Black 5 azo dye was created, based on PbO$_2$-CTAB/g-C$_3$N$_4$ anode material as an electrocatalyst. The parameters that were optimized include initial pH, concentration of $Na_2SO_4$ as supporting electrolyte, electric current, and concentration of RB5 dye. PbO$_2$ probes synthesized using different surfactants as templates were used separately in pristine form and as composite materials with graphitic carbon nitride. Morphological and electrochemical properties of the materials were evaluated using PXRD, TEM, SEM, CV, and EIS techniques. The efficiency of the electrolytic reaction vastly depended on procedure parameters. The results showed that PbO$_2$-CTAB/g-C$_3$N$_4$ exhibits the best electrocatalytic properties and the highest removal rate for RB5, with 90% of the dye removed after 60 min of treatment. The specific energy consumption for this process was 0.4374 kWh/g. Finally, the prepared material showed good stability and enabled the treatment of highly polluted wastewaters with excellent results, but also, due to its electrocatalytic properties, could be applied in various fields of electrochemistry, such as sensing and biosensing probes. With regard to efficiency and longevity, the obtained results suggest that the proposed method can offer a low-cost, effective, and green approach in the field of environmental control, and due to its simplicity, this method has high potential for technology transfer.

**Author Contributions:** Conceptualization, A.M., M.O., A.K., S.S. and D.M.S.; methodology, investigation, A.M., S.S., A.K., M.O. and D.M.; resources, M.O. and D.M.S.; writing—original draft preparation, A.M., S.S., M.O. and D.M.S.; writing—review and editing, A.M., S.S., M.O. and D.M.S.; supervision, D.M.S.; project administration, Z.K., D.M. and D.M.S.; funding acquisition, D.M.S. All authors have read and agreed to the published version of the manuscript.

**Funding:** This work was supported by Ministry of Education, Science and Technological Development of Republic of Serbia Contract number: 451-03-68/2022-14/200168, EUREKA project E!13303 and Ministry of Science and Higher Education of the Russian Federation (agreement No. 075-15-2022-1135) and South Ural State University.

**Data Availability Statement:** The study did not report any data.

**Conflicts of Interest:** The authors declare that they have no known competing financial interest or personal relationship that could have appeared to influence the work reported in this paper.

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
