# Peer review of "Differently Prepared PbO2/Graphitic Carbon Nitride Composites for Efficient Electrochemical Removal of Reactive Black 5 Dye"

_catalysts, doi:10.3390/catal13020328_

Round 1
Reviewer 1 Report
The authors have presented differently prepared PbO2/graphitic carbon nitride composites to effectively remove Reactive Black 5 dye via electrochemical oxidation.
Manuscript lacks novelty. There are numerous reports promising similar studies so what’s the novelty of the work? Whole draft needs English editing. In present form, it’s difficult to understand phrases.
State of the art is completely lacking in Introduction section. Currently, introduction is like description RB5 dyes and other methods used in literature to degrade RB5. Please make clear justification about using your methodology. What about others electrochemical oxidation used in literature for Degradation of Reactive Black 5? How is your method better or superior? I can’t find any literature cited about using electrochemical oxidation for dye degradtion
There is no comparison data reported anywhere, which confirms used technique is efficient or comparable with other technique in literature.
Line 60-62 “Electrochemical oxidation processes could be effective logical choice for the develop ment of green, time-effective and at the same time potent methods for the removal of various pollutants, including reactive textile dyes [19–21]”. How electrochemical oxidation process can be green? Elaborate! Also, how used method is better than biodegradation and bio catalysis, photocatalysis, electrocoagulation etc. is not clear. Please elaborate in detail.
Line 26-27 Wavelength range is missing..
Why there is question mark in figure 3 legend. Please remove.
Line 43-45 Doesn’t make any sense. Please re-write.
Line 49 typo error…..
Line 52-53 Not clear. Please re-write.
Line 58-59 please re-write.
Figure 3A and 3B, needs to re-draw again. Fonts are not visible clearly; Figure legend are cut at top (3A).
Line 138-160 Very difficult to follow the figure with text. Please write each sentence clearly.
Line 177 full stop is missing.
Justification of getting lower degradation rate for 0.2 M Na2SO4 electrolyte is not clear.
What about considering ph. less than 2 to get better degradation of dye? Scientific explanation is missing, why highest degradation is obtained at ph=2.
Provide detail mechanism about degradation of RB5 dye via used electrochemical oxidation.
Authors should provide a schematic path to show RB5 dye (which is present), along the possible structure of the degradation products for proper understanding to readers.
Also provide evidence for identification of by-products?
Lots of spelling mistake and typo errors in whole manuscript.
All figures are very poorly displayed, figure captions are incomplete and needs to re-write again. Authors should provide the figures in high resolution images.
Please rewrite the Abstract. In general, the abstract is a condensation of the information (facts) in the paper (150-250 words) and should present briefly and clearly the methods of the research and the principal results
The Conclusions section is not sound. It should include the major conclusions, the limitations of the work and the future work.
The manuscript needs thorough revision to improve the text quality and readability of the work. Currently, whole draft is written like lab report. In current form, it should be rejected. Further, systematic explanation of experimental procedure and scientific interpretation of the obtained results should be presented.
Author Response
Dear Editor,
Thank you so much for giving us an opportunity to revise our manuscript, and thus improveour work according to the reviewers’ valuable comments. We have carefully considered all reviewers’ questions and all the weak points of our work and have significantly changed ourmanuscript accordingly. The list of changes has been written below, and inserted changes inthe text of the re-submitted manuscript are being highlighted. We hope that the revised version of the manuscript will satisfy yours and the reviewers’ requirements for publication in the Journal.
Reviewers’ comments:
Reviewer 1.
We are grateful to the reviewer for their time and efforts and positive comments of our work. We revised manuscript according to these comments and we hope that this improvement, based on reviewer comments will significantly improve final version of our work.
The authors have presented differently prepared PbO2/graphitic carbon nitride composites to effectively remove Reactive Black 5 dye via electrochemical oxidation.
Q1 Manuscript lacks novelty. There are numerous reports promising similar studies so what’s the novelty of the work? Whole draft needs English editing. In present form, it’s difficult to understand phrases. State of the art is completely lacking in Introduction section. Currently, introduction is like description RB5 dyes and other methods used in literature to degrade RB5. Please make clear justification about using your methodology. What about others electrochemical oxidation used in literature for Degradation of Reactive Black 5? How is your method better or superior? I can’t find any literature cited about using electrochemical oxidation for dye degradation
A1 We thank the reviewer for their insightful comments and suggestions. The introduction has now been expanded with a wider overview of previous works and provides a justification for choosing the materials and methods we wanted to study here. The comparison with other methods is now given under results and discussions (see A2).
Electrochemical oxidation processes present an effective and logical choice for the development of green, time-effective and, at the same time, potent methods for the removal of various pollutants, including reactive textile dyes. [1–3] Electrocatalytic processes can be considered green primarily due to mild and environmentally acceptable working conditions (ambient pressure and temperature, work in aqueous solutions, no additional chemicals or, when they are used, they have lower toxicity). These methods may also be coupled with renewable energy sources, [4] they do not require systems for temperature or pressure control nor expensive gases and give the possibility of avoiding expensive catalysts based on precious metals, so their potential for large-scale application is enormous.
Therefore, electrochemical methods, through the rational design of the electrocatalytic setup, enable simple and practical systems for water purification and pollutant removal. Our research group proposed several approaches for the efficient removal of organic pollutants using electrooxidation methods, such as for the removal of Reactive Blue 52, [5,6] triketone herbicides [7] or ibuprofen. [8]
For electrochemical oxidative processes, anodes with high oxygen evolution potentials (non-active anodes) achieve complete mineralization of organics and are thus favored for wastewater remediation. The most commonly used non-active anode is the boron doped diamond electrode, but cheaper alternatives include various metal oxides such as TiO2, SnO2 and PbO2. [9] Metal oxides exhibit exceptional properties valuable to electrochemical processes, while featuring reduced costs, availability and environmental compatibility, and have thus found diverse applications. The properties of metal oxides can be fine-tuned by controlling their morphology, size and crystallinity, which can be achieved using template synthesis.
Lead dioxide is an inexpensive and widely available material, readily incorporated in composite materials with outstanding properties. [10] Lead oxide-based anodes gave promising performance for electrochemical degradation of organic pollutants wastewater treatment [11], as they can possess unique catalytically active surfaces, [12] good stability and a long electrode life. [13]
Graphitic carbon nitride (g-C3N4) has found numerous applications owing to its properties, such as high surface area, low cost and biocompatibility. The use of g-C3N4 in composites proved to enhance its photo- and electrocatalytic properties. Good results for the photocatalytic degradation of organic pollutants were achieved when using nanocomposites made from metal oxides and g-C3N4. [14–19] On the other hand, reports of g-C3N4 being used for water remediation in a purely electrochemical setting are scarce.
Q2 There is no comparison data reported anywhere, which confirms used technique is efficient or comparable with other technique in literature.
A2 Thank you for your insight, we included the comparison with other methods under Results and Discussions.
Table 2 provides and overview of different electrochemical systems employed to remove RB5, with focus on results with Na2SO4 as supporting electrolyte. All selected studies reported faster RB5 removal when using NaCl or KBr [20–23], but the use of these non-inert salts is discouraged in electrochemistry, because they lead to the formation of chlorinated and brominated degradation products, which are often more toxic compared to the parent compound. [22,24,25]
When excluding the results with NaCl, our study gave the fastest (under 1 h) and the highest RB5 removal (nearly 90%). The system with Ti/CoOx–RuO2–SnO2–Sb2O5 led removed only 40% of the selected dye, during 2 h of treatment in the presence of Na2SO4, but increased to over 95% when NaCl was used. [21] The next study with Ti/SnO2-Sb2O5-IrO2 was slightly improved in the same time interval without the addition of NaCl. [23] Graphite electrode lead to complete RB5 degradation and quite efficient COD removal, but this experimental setup lasted for three hours. [22]
Table 2. The RB5 degradation efficiency comparison with the literature data
|
Material |
Supporting electrolyte |
% of RB5 removal and time |
Energy consumption |
Ref |
|
RuO2/IrO2/TiO2@DSA® |
0.008 M NaCl |
~100% in 15 min |
Not reported |
[20] |
|
Ti/CoOx–RuO2–SnO2–Sb2O5 |
0.07 M Na2SO4 |
~40% in 2 h |
34.5 kWh/kg |
[21] |
|
Ti/SnO2-Sb2O5-IrO2 |
0.1 M Na2SO4 |
~60% in 2 h |
Not reported |
[23] |
|
Graphite |
0.1 M Na2SO4 |
~100% in 3 h |
Not reported |
[22] |
|
PbO2-CTAB/g-C3N4 on SS |
0.15 M Na2SO4 |
~90 % in 1 h |
0.4374 kWh/g |
This study |
a calculated as general current efficiency, based on COD removal. [26]
Q3 Line 60-62 “Electrochemical oxidation processes could be effective logical choice for the development of green, time-effective and at the same time potent methods for the removal of various pollutants, including reactive textile dyes [19–21]”. How electrochemical oxidation process can be green? Elaborate! Also, how used method is better than biodegradation and bio catalysis, photocatalysis, electrocoagulation etc. is not clear. Please elaborate in detail.
A3 Thank you very much for your question. In the revised version we provided additional explanation for this statement. This is highlighted in the revised manuscript.
Electrochemical oxidation processes present an effective and logical choice for the development of green, time-effective and, at the same time, potent methods for the removal of various pollutants, including reactive textile dyes. [1–3] Electrocatalytic processes can be considered green primarily due to mild and environmentally acceptable working conditions (ambient pressure and temperature, work in aqueous solutions, no additional chemicals or, when they are used, they have lower toxicity). These methods may also be coupled with renewable energy sources, [4] they do not require systems for temperature or pressure control nor expensive gases and give the possibility of avoiding expensive catalysts based on precious metals, so their potential for large-scale application is enormous.
Therefore, electrochemical methods, through the rational design of the electrocatalytic setup, enable simple and practical systems for water purification and pollutant removal.
Q4 Line 26-27 Wavelength range is missing.
A4 We have concluded that this sentence is redundant in the abstract so it has been removed.
Q5 Why there is question mark in figure 3 legend. Please remove.
A5 Thank you for this remark. This was technical mistake. In the revised version we provided new Figure 3 and appropriate Figure legend.
Figure 3. (A) RB5 removal under pH 2, 4, and 6; (B) UV-Vis spectra of RB5 solution in the range from 200 to 800 nm during 60 minutes of treatment (pH 2, 0.1 M Na2SO4, 5V) at different time intervals
Q6 Line 43-45 Doesn’t make any sense. Please re-write.
A6 Thank you for your comment. Corrected.
The release of colored wastewaters is not only aesthetically unpleasant, but also obstructs penetration of light, which affects biological processes. It can also potentially generate toxicity for aquatic organisms and finally, via food chain or exposure, in humans [5–9].
Q7 Line 49 typo error.
A7 Thank you for your comment. Corrected.
Q8 Line 52-53 Not clear. Please re-write.
A8 Thank you for your comment. After careful consideration of this part of the manuscript, we decided to exclude this reference.
Q9 Line 58-59 please re-write.
A9 Thank you for your insight, this part is now revised.
Biodegradation can yield very good results but it can also be less effective than other methods, because dyes can be toxic for bacteria and can thus inhibit their activity.
Q10 Figure 3A and 3B, needs to re-draw again. Fonts are not visible clearly; Figure legend are cut at top (3A).
A10 Thank you for this comment. In the revised version we provided new Figure 3 with all the required changes.
Figure 3. (A) RB5 removal under pH 2, 4, and 6; (B) UV-Vis spectra of RB5 solution in the range from 200 to 800 nm during 60 minutes of treatment (pH 2, 0.1 M Na2SO4, 5V) at different time intervals
Q11 Line 138-160 Very difficult to follow the figure with text. Please write each sentence clearly.
A11 Thank you for your suggestion. In the revised version we improved English language deeply by native English speaker. This part of the manuscript is now rewritten and more clearly explained.
2.2.1. pH optimization
Starting pH of the supporting solution represents one of the most important factors for the performance of electrochemical processes. In order to determine the optimal initial pH for electrochemical degradation, RB5 was dissolved in 0.1 M Na2SO4 to make a 70 mg/L solution with native pH 5.63 (measured using pH meter). The λmax of this solution obtained by the spectrometer was 598 nm. Three initial pH values were tested – 2, 4 and 6. The native pH of the stock solution was adjusted to these values using 0.1 M NaOH and 0.1 M H2SO4. The electrochemical reaction was allowed to continue almost to the point when the absorbance on the UV/Vis spectrum reached a plateau. For the reaction, one stainless steel (SS) electrode (4 cm2 of surface area) modified with PbO2-CTAB/g-C3N4 was used as the anode while an unmodified SS electrode was used as the cathode. The experiments were performed under constant a voltage of 5 V and the results are shown in Figure 3.
Figure 3. (A) RB5 removal under pH 2, 4, and 6; (B) UV-Vis spectra of RB5 solution in the range from 200 to 800 nm during 60 minutes of treatment (pH 2, 0.1 M Na2SO4, 5V) at different time intervals
As can be seen in Figure 3A, the best degradation rate was observed at initial pH 2. After 60 minutes of treatment using this starting pH, 98% of the color was removed. This can be attributed to the cleavage of the –N=N– bonds and aromatic rings, resulting in the decrease of optical density of the dye solution (Figure 3B).
Also, in the acidic medium, OH• radicals are produced by the anodic discharge of water in the indirect electrochemical oxidation of organic dyes at the anode. These OH• radicals adsorb onto the anode surface and oxidize the organic material. At higher pH, it could be expected that a larger amount of hydroxyl radicals would be generated, which would result in a stronger efficiency of the system. However, it was determined that the degree of ionization of the cationic dye is highly dependent on the initial pH value of the solution, which most often leads to the formation of a precipitated photochromic compound and a decrease in resistance on mass transfer at higher pH. Therefore, an acidic environment leads to better dissolution and degree of ionization. In the case of an increase in the pH value of the solution, an increased consumption of electrolytes occurs dominantly, which directly affects the conductivity of the solution. The lower pH values were not tested as recent studies showed that low pH values led to a higher dissolution rate of lead dioxide [32] and that these values are not appropriate for work with lead dioxide films, as they can cause the formation of lead sulfate. [33] The occurrence of new peaks in the spectrum confirms the formation of smaller molecules. The increase in absorbance for these peaks was followed by the decrease of the main RB5 peak, suggesting a relationship between these processes and successful decomposition of RB5. Thus, based on the conducted study, all further experiments were done using starting pH 2 for the dye solutions.
___________________________________________________________________________
Q12 Line 177 full stop is missing.
A12 Thank you for noticing. Corrected.
Q13 Justification of getting lower degradation rate for 0.2 M Na2SO4 electrolyte is not clear.
A13 Thank you for this remark. In the revised version we provided additional explanation for this, supported by appropriate reference.
Belal and coworkers studied the effect of the supporting electrolytes and their concentrations on the dye removal efficiency and found a similar trend. [34] The decrease in the removal rate can be connected with the generation of fewer antioxidants at higher concentrations, as a result of the deacceleration of the important reactive species for the degradation pathway, due to increased production of S2O82- from sulfate ions.
Q14 What about considering ph. less than 2 to get better degradation of dye? Scientific explanation is missing; why highest degradation is obtained at ph=2.
A14 Thank you for the comment. We agree with the reviewer that current explanations are not sufficiently clear. In the revised version we provided additional explanation regarding effect of the pH on the degradation rate.
Also, in the acidic medium, OH• radicals are produced by the anodic discharge of water in the indirect electrochemical oxidation of organic dyes at the anode. These OH• radicals adsorb onto the anode surface and oxidize the organic material. At higher pH, it could be expected that a larger amount of hydroxyl radicals would be generated, which would result in a stronger efficiency of the system. However, it was determined that the degree of ionization of the cationic dye is highly dependent on the initial pH value of the solution, which most often leads to the formation of a precipitated photochromic compound and a decrease in resistance on mass transfer at higher pH. Therefore, an acidic environment leads to better dissolution and degree of ionization. In the case of an increase in the pH value of the solution, an increased consumption of electrolytes occurs dominantly, which directly affects the conductivity of the solution. The lower pH values were not tested as recent studies showed that low pH values led to a higher dissolution rate of lead dioxide [32] and that these values are not appropriate for work with lead dioxide films, as they can cause the formation of lead sulfate. [33]
Q15 Provide detail mechanism about degradation of RB5 dye via used electrochemical oxidation. Authors should provide a schematic path to show RB5 dye (which is present), along the possible structure of the degradation products for proper understanding to readers. Also provide evidence for identification of by-products?
A15 Thank you for your comment. Our initial idea was to determine the degradation products using LC-MS, but we could not develop a reliable method for liquid chromatography with a DAD detector, so we decided to stick with only UV-Vis data. Please, find the samples of the chromatographs in Fig. 1. A (standards) and B (samples of RB5 treated solution using PbO2-CTAB-g-C3N4, including starting solution, and samples after 30, 60, 90, 120, and 180 min of electrochemical oxidation). As can be seen, there is no constant retention time, and even though there is clear evidence of RB5 degradation, we did not feel confident enough to present these results and continue with the LC-MS results.
Used method an sample preparattion: HPLC-DAD method. Prior to analysis, samples and standard solutions were filtered using Syringe filters (PTFE membrane, 0.45 µm, 25 mm, Agilent Technologies. The HPLC-DAD system applied for the RB5 concentration monitoring was Thermo Ultimate 3000 RS, ThermoFisher Scientific, Germany. Five μL of samples were injected into the Hypersil Gold aQ C18 (150 mm × 3 mm, 3 μm, ThermoFisher Scientific, Germany) at 25 °C and eluted in gradient mode using water (eluent A, HPLC grade) and acetonitrile (eluent B, HPLC grade). The analysis lasted for 10 minutes, starting with 98% of B and keeping in this ratio for 1 minute, then dropping % of B to 50% during the next two minutes, followed by 58% of B in the 5th minute and ending with increasing % of B to starting conditions for the rest of the method time. [35] The DAD detector was set at 254, 310, 392, and 598 nm, while the instrument was controlled and data collected using the Chromeleon 6.8 (ThermoFisher Scientific, Germany) software.
Fig. 1. Chromatograms of standard solutions of RB5 (A), ranging 2-100 mg/dm3 and samples of RB5 treated solution using PbO2-CTAB- g-C3N4 (B), monitored on 254 nm using HPLC-DAD.
Instead, we provided a theoretically obtained degradation products. If the reviewer or editor decide that this approach is acceptable, we would gladly incorporate this method and results into the manuscript as a new paragraph: Theoretical insights into reaction mechanism.
Computational details and theoretically obtained results:
All quantum chemical calculations were carried out with Gaussian 16 [36] electronic structure program suite (Revision A.03), by using Density Functional Theory (DFT) [37] approach. The well-known and widely used B3LYP [38,39] functional, together with 6-31+G [40–43] orbital basis set and empirical GD3BJ dispersion correction [44] has been chosen for calculation of structural and electronic properties of the studied dye RB5. To make calculations as realistic as possible, solvation effects of water have been included using the polarizable conductor continuum model (C-PCM) trough solvent cavity reaction field (SCRF) method [45–47]. Investigated molecule was fully optimized within the framework of described computational conditions. Mulliken [48] charge analysis information, together with the condensed forms of the Fukui [49–52] functions (and corresponding equations), for investigated RB5 and all related ionized forms of RB5 are provided in Table 1.
The condensed forms of the Fukui function for an atom j in a specific molecule, are calculated using the equations 1 - 3, for a nucleophilic electrophilic , and radical attack respectively. In these equations qj is the atomic charge, evaluated from the Mulliken charge analysis, at the jth atomic site in the neutral (N), anionic (N+1), or cationic (N-1) chemical species. N stands for fully optimized RB5 molecule, and N+1 and N-1 represent corresponding nonoptimized anionic and cationic form (It is important to keep in mind that all ionizable functional groups of RB5, in working experimental (acidic) conditions, are protonated (Figure 2). For this reason, investigated molecule is in cationic form, thus, one-electron donation will generate a neutral structure. Following the same analogy, after one electron removal a di-cation adduct is formed.
, (1)
, (2)
(3)
Most probable reaction sites, based on these calculations, are shown in Figure 2 and calculated values are summarized in Table 1 (since hydrogen atoms doesn’t contain any important insight, in order to be as concise as possible, we have provided only the values for the heavy atoms). Abbreviations Nu. El. and Rad. represent reactivity towards nucleophiles, electrophiles and radical species respectively and are graphically shown in the Figure 2 as green, blue and red cycles, respectively. Orbital analysis was systematically undertaken to view the localization and shape of highest occupied molecular orbital (HOMO), lowest unoccupied molecular orbital (LUMO), as well as the spin-density, formed upon one-electron oxidation.
Figure 2. a.) The structure of investigated RB5 in working (acidic) conditions. 2.) HOMO and LUMO orbitals c.) spin-density, generated upon one-electron oxidation process c.) Most probable reaction centers based on calculated Fukui functions (towards nucleophilic – green, electrophilic –blue, and radical attack - red); calculations carried out on B3LYP-D3BJ level of theory
Figure 3. Proposed degradation mechanism of investigated RB5, based on our theoretically obtained insights
In order to gain more information about the degradation mechanism, we firstly inspected the location (distribution) and shape of HOMO and LUMO orbitals, since these orbitals are directly correlated with electrochemical events. The shape and distribution of HOMO orbital clearly reveals larger investment of the hydroxyl group and thus the aromatic ring on which this group is located (Figure 2). For this reason, upon electrochemical oxidation of investigated chemical moiety, formed electron density is dominantly concentrated in this region (Figure 3). Considerable amount of generated spin density is accumulated in the neighboring diazo substructure, which can cause a chemical change and possible hydrolysis in this region. Upon initial oxidation, RB5 will become more reactive, and as such prone to the attack of reactive species, present in the solution. Although we shall not neglect the possibility of electrophilic attack, our analyte is in its oxidized form, thus we can expect it to be more reactive towards nucleophilic (water molecules, present in the solution) and radical attacks (hydroxyl radicals, formed on the electrode surface). Most reactive chemical substructures, with the most prominent Fukui functions for the nucleophilic and electrophilic attack (Table 1), are nitrogen atoms 15 and 16, complementary sulfur atoms 3 and 4, 65 and 67, a and finally aromatic carbon atoms 34 and 35. Upon nucleophilic/electrophilic attack of reactive species, present in the solution, we can expect a chemical alteration of affected centers (and further oxidation), and thus hydrolysis of periphery sulfonyl groups, as well as diazo bonds, connecting the branches to the aromatic core. It is important to notice that the diazo substructure neighboring the hydroxyl group shows higher affinity towards nucleophilic/electrophilic attack, thus this kind of reaction will probably be the main cause of chemical change. Final reactive centers, the aromatic carbon atoms, will most likely undergo further chemical change (oxidation) into phenols/quinones. The region shoving the highest affinity towards radical attack is the diazo substructure neighboring the amine group, thus we can expect the radical attack to be responsible for chemical change and further hydrolysis formed product. Our calculations also revealed that enhanced reactivity of oxygen 5, and since we already mentioned that the initial oxidation takes place in the region of hydroxyl group, we can expect some reactive hybrid structures, with shared electron density between neighboring oxygen 5 and nitrogen 85, to be generated, with the final termination products in the form of para-quinone derivatives.
Considered together, our results lead us to conclude that after initial oxidation of RB5, reactive cationic moiety undergoes many different chemical changes (reactions) and final hydrolysis, but in order to gain the precise knowledge about the order and magnitude of these changes, more theoretical work, together with experimental justification must be provided.
Table 1. Mulliken charge analysis and calculated Fukui functions for investigated RB5
|
No. |
Atom |
N |
N-1 |
N+1 |
nu |
el |
rad |
|
1 |
S |
1.547 |
1.477 |
1.476 |
-0.1 |
0.1 |
0.0 |
|
2 |
S |
1.510 |
1.382 |
1.384 |
-0.1 |
0.1 |
0.0 |
|
3 |
S |
1.240 |
0.959 |
0.959 |
-0.3 |
0.3 |
0.0 |
|
4 |
S |
1.292 |
0.974 |
0.973 |
-0.3 |
0.3 |
0.0 |
|
5 |
O |
-0.570 |
-0.480 |
-0.596 |
0.0 |
-0.1 |
-0.1 |
|
6 |
O |
-0.417 |
-0.356 |
-0.395 |
0.0 |
-0.1 |
0.0 |
|
7 |
O |
-0.436 |
-0.382 |
-0.423 |
0.0 |
-0.1 |
0.0 |
|
8 |
O |
-0.548 |
-0.526 |
-0.566 |
0.0 |
0.0 |
0.0 |
|
9 |
O |
-0.546 |
-0.517 |
-0.555 |
0.0 |
0.0 |
0.0 |
|
10 |
O |
-0.547 |
-0.511 |
-0.535 |
0.0 |
0.0 |
0.0 |
|
11 |
O |
-0.531 |
-0.521 |
-0.546 |
0.0 |
0.0 |
0.0 |
|
12 |
O |
-0.547 |
-0.502 |
-0.521 |
0.0 |
0.0 |
0.0 |
|
13 |
O |
-0.541 |
-0.515 |
-0.535 |
0.0 |
0.0 |
0.0 |
|
14 |
N |
-0.362 |
-0.149 |
-0.252 |
0.1 |
-0.2 |
-0.1 |
|
15 |
N |
-0.606 |
-0.310 |
-0.352 |
0.3 |
-0.3 |
0.0 |
|
16 |
N |
0.334 |
0.181 |
-0.049 |
-0.4 |
0.2 |
-0.1 |
|
17 |
N |
0.471 |
0.312 |
0.190 |
-0.3 |
0.2 |
-0.1 |
|
18 |
C |
0.071 |
0.240 |
0.130 |
0.1 |
-0.2 |
-0.1 |
|
19 |
C |
-0.204 |
-0.163 |
-0.222 |
0.0 |
0.0 |
0.0 |
|
20 |
C |
0.240 |
0.352 |
0.276 |
0.0 |
-0.1 |
0.0 |
|
21 |
C |
-1.042 |
-0.885 |
-0.969 |
0.1 |
-0.2 |
0.0 |
|
22 |
C |
-0.239 |
-0.351 |
-0.339 |
-0.1 |
0.1 |
0.0 |
|
23 |
C |
0.508 |
0.404 |
0.385 |
-0.1 |
0.1 |
0.0 |
|
24 |
C |
-0.945 |
-0.787 |
-0.829 |
0.1 |
-0.2 |
0.0 |
|
25 |
C |
-0.148 |
-0.121 |
-0.131 |
0.0 |
0.0 |
0.0 |
|
26 |
C |
0.290 |
0.236 |
0.257 |
0.0 |
0.1 |
0.0 |
|
27 |
C |
0.233 |
0.140 |
0.095 |
-0.1 |
0.1 |
0.0 |
|
28 |
C |
-0.532 |
-0.630 |
-0.458 |
0.1 |
0.1 |
0.1 |
|
29 |
C |
-0.611 |
-0.501 |
-0.434 |
0.2 |
-0.1 |
0.0 |
|
30 |
C |
-0.011 |
0.122 |
0.045 |
0.1 |
-0.1 |
0.0 |
|
31 |
C |
0.110 |
0.148 |
0.084 |
0.0 |
0.0 |
0.0 |
|
32 |
C |
0.019 |
-0.141 |
-0.208 |
-0.2 |
0.2 |
0.0 |
|
33 |
C |
-0.226 |
-0.023 |
-0.173 |
0.1 |
-0.2 |
-0.1 |
|
34 |
C |
-0.219 |
0.139 |
0.053 |
0.3 |
-0.4 |
0.0 |
|
35 |
C |
0.061 |
-0.341 |
-0.372 |
-0.4 |
0.4 |
0.0 |
|
36 |
C |
-0.058 |
-0.114 |
-0.152 |
-0.1 |
0.1 |
0.0 |
|
37 |
C |
-0.320 |
-0.144 |
-0.134 |
0.2 |
-0.2 |
0.0 |
|
38 |
C |
-0.280 |
-0.177 |
-0.175 |
0.1 |
-0.1 |
0.0 |
|
39 |
C |
-0.245 |
-0.138 |
-0.146 |
0.1 |
-0.1 |
0.0 |
|
51 |
C |
-0.432 |
-0.370 |
-0.367 |
0.1 |
-0.1 |
0.0 |
|
54 |
C |
-0.509 |
-0.557 |
-0.561 |
-0.1 |
0.0 |
0.0 |
|
57 |
C |
-0.433 |
-0.368 |
-0.362 |
0.1 |
-0.1 |
0.0 |
|
60 |
C |
-0.499 |
-0.557 |
-0.567 |
-0.1 |
0.1 |
0.0 |
|
63 |
O |
-0.406 |
-0.306 |
-0.305 |
0.1 |
-0.1 |
0.0 |
|
64 |
O |
-0.407 |
-0.306 |
-0.306 |
0.1 |
-0.1 |
0.0 |
|
65 |
S |
1.512 |
1.277 |
1.275 |
-0.2 |
0.2 |
0.0 |
|
66 |
O |
-0.559 |
-0.495 |
-0.497 |
0.1 |
-0.1 |
0.0 |
|
67 |
S |
1.504 |
1.276 |
1.272 |
-0.2 |
0.2 |
0.0 |
|
68 |
O |
-0.556 |
-0.496 |
-0.502 |
0.1 |
-0.1 |
0.0 |
|
69 |
O |
-0.578 |
-0.504 |
-0.506 |
0.1 |
-0.1 |
0.0 |
|
70 |
O |
-0.575 |
-0.505 |
-0.511 |
0.1 |
-0.1 |
0.0 |
|
71 |
O |
-0.633 |
-0.565 |
-0.566 |
0.1 |
-0.1 |
0.0 |
|
73 |
O |
-0.633 |
-0.565 |
-0.570 |
0.1 |
-0.1 |
0.0 |
|
75 |
O |
-0.493 |
-0.494 |
-0.523 |
0.0 |
0.0 |
0.0 |
|
77 |
O |
-0.506 |
-0.495 |
-0.525 |
0.0 |
0.0 |
0.0 |
|
82 |
N |
-1.099 |
-1.062 |
-1.062 |
0.0 |
0.0 |
0.0 |
Q16 Lots of spelling mistake and typo errors in whole manuscript.
A16 Thank you for your suggestion. The entire manuscript has undergone an extensive revision in orthography, grammar and clarity of text.
Q17 All figures are very poorly displayed, figure captions are incomplete and needs to re-write again. Authors should provide the figures in high resolution images.
A17 Thank you for your suggestion. Improvement was suggested from other reviewers too. The figures were changed, necessary improvements were done and where suggested new figures were added.
Q18 Please rewrite the Abstract. In general, the abstract is a condensation of the information (facts) in the paper (150-250 words) and should present briefly and clearly the methods of the research and the principal results.
A18 Thank you for your insight, we revised the abstract to reflect the proposed method of RB5 degradation.
In this paper, electrochemical degradation of Reactive Black 5 (RB5) textile azo dye was examined in regard to different synthesis procedures for making PbO2 – graphitic carbon nitride (g-C3N4) electrode. The reaction of with ClO− in the presence of different surfactants, i.e., cetyltrimethylammonium bromide (CTAB) and tetrabutylammonium phosphate (TBAP), under conventional conditions, resulted in the formation of PbO2 with varying morphology. The obtained materials were combined with g-C3N4 for the preparation of the final composite materials, which were then characterized morphologically and electrochemically. After optimizing the degradation method, it was shown that an anode comprising a steel electrode coated with the composite of PbO2 synthesized using CTAB as template and g-C3N4, and using 0.15 M Na2SO4 as the supporting electrolyte, gave the best performance for RB5 dye removal from a 35 mg/L solution. The treatment duration was 60 minutes, applying a current of 0.17 A (electrode surface 4 cm2, current density of 42.5 mA/cm2), while the initial pH of the testing solution was 2. The reusability and longevity of the electrode surface (which showed no significant change in activity throughout the study) may suggest that this approach is a promising candidate for wastewater treatment and pollutant removal.
Q19 The Conclusions section is not sound. It should include the major conclusions, the limitations of the work and the future work.
A19 Thank you for this suggestion. This was also commented from the reviewer 2. In the revised version we rewrote section Conclusions. This is highlighted in the revised manuscript.
In this paper, an electrochemical degradation/removal method for Reactive Black 5 azo dye was created, based on PbO2-CTAB/g-C3N4 anode material as an electrocatalyst. The parameters that were optimized include initial pH, concentration of Na2SO4 as supporting electrolyte, electric current and concentration of RB5 dye. PbO2 probes synthesized using different surfactants as templates were used separately in pristine form and as composite materials with graphitic carbon nitride. Morphological and electrochemical properties of the materials were evaluated using PXRD, TEM, SEM, CV and EIS techniques. The efficiency of the electrolytic reaction vastly depended on procedure parameters. The results showed that PbO2-CTAB/g-C3N4 exhibits the best electrocatalytic properties and the highest removal rate for RB5, with 90% of the dye removed after 60 minutes of treatment. The specific energy consumption for this process was 0.4374 kWh/g. Finally, the prepared material showed good stability and enabled the treatment of highly polluted wastewaters with excellent results, but also, due to its electrocatalytic properties, could be applied in various fields of electrochemistry, such as sensing and biosensing probes. In regards to efficiency and longevity, the obtained results suggest that the proposed method can offer a low-cost, effective and green approach in the field of environmental control, and due to its simplicity, this method has high potential for technology transfer.
Q20 The manuscript needs thorough revision to improve the text quality and readability of the work. Currently, whole draft is written like lab report. In current form, it should be rejected. Further, systematic explanation of experimental procedure and scientific interpretation of the obtained results should be presented.
A20 We are grateful to the reviewer for their time and efforts and constructive criticism of our paper. We revised manuscript according to these comments and we hope that this improvement will significantly improve final version of our work.
Reviewer 2
Very interesting and well organized study. However, there are a number of comments that I hope will undoubtedly improve this manuscript:
Q1 Authors need to revise the list of keywords and add more specific terms directly related to this study.
A1 Thank you for your insight, we revised keywords to better reflect this article.
reactive azo dye; surfactant-assisted synthesis; electrode morphology; advanced oxidation pro-cesses; lead dioxide; energy efficiency
Q2 The introduction section should be revised and more clearly demonstrate the relevance and novelty of the study.
A2 The introduction has been revised and further expanded so as to give a short comprehensive review of the advantages and disadvantages of different methods for wastewater remediation, thus establishing the relevance of this study. The novelty of this work involves the syntheses of new materials inspired by previous successful studies and employing them in an original method for the degradation of organic pollutants, using RB5 as a model compound.
Each of these methods has its positive and negative sides. For instance, electrocoagulation, membrane separation processes, adsorption and precipitation only change the phase of pollutants. Photo- and chemical oxidation require additional chemicals and oxidation agents that are considered highly toxic and can produce additional hazardous waste. Photochemical oxidation is one of the most commonly used methods for the degradation of waterborne pollutants. [53] TiO2-based materials are most widely used because of their unprecedented photocatalytic activity, but other metallic and nonmetallic catalysts are giving promising results. [54–56] Biodegradation can yield very good results but it can also be less effective than other methods, because dyes can be toxic for bacteria and can thus inhibit their activity.
Electrochemical oxidation processes present an effective and logical choice for the development of green, time-effective and, at the same time, potent methods for the removal of various pollutants, including reactive textile dyes. [1–3] Electrocatalytic processes can be considered green primarily due to mild and environmentally acceptable working conditions (ambient pressure and temperature, work in aqueous solutions, no additional chemicals or, when they are used, they have lower toxicity). These methods may also be coupled with renewable energy sources, [4] they do not require systems for temperature or pressure control nor expensive gases and give the possibility of avoiding expensive catalysts based on precious metals, so their potential for large-scale application is enormous.
Therefore, electrochemical methods, through the rational design of the electrocatalytic setup, enable simple and practical systems for water purification and pollutant removal. Our research group proposed several approaches for the efficient removal of organic pollutants using electrooxidation methods, such as for the removal of Reactive Blue 52, [5,6] triketone herbicides [7] or ibuprofen. [8]
For electrochemical oxidative processes, anodes with high oxygen evolution potentials (non-active anodes) achieve complete mineralization of organics and are thus favored for wastewater remediation. The most commonly used non-active anode is the boron doped diamond electrode, but cheaper alternatives include various metal oxides such as TiO2, SnO2 and PbO2. [9] Metal oxides exhibit exceptional properties valuable to electrochemical processes, while featuring reduced costs, availability and environmental compatibility, and have thus found diverse applications. The properties of metal oxides can be fine-tuned by controlling their morphology, size and crystallinity, which can be achieved using template synthesis.
Lead dioxide is an inexpensive and widely available material, readily incorporated in composite materials with outstanding properties. [10] Lead oxide-based anodes gave promising performance for electrochemical degradation of organic pollutants wastewater treatment [11], as they can possess unique catalytically active surfaces, [12] good stability and a long electrode life. [13]
Graphitic carbon nitride (g-C3N4) has found numerous applications owing to its properties, such as high surface area, low cost and biocompatibility. The use of g-C3N4 in composites proved to enhance its photo- and electrocatalytic properties. Good results for the photocatalytic degradation of organic pollutants were achieved when using nanocomposites made from metal oxides and g-C3N4. [14–19] On the other hand, reports of g-C3N4 being used for water remediation in a purely electrochemical setting are scarce.
Q3 In lines 54-64, the authors describe the shortcomings of the main methods for removing organic dyes. However, one of the most widely popular and used techniques - photocatalysis, was mentioned very superficially. Meanwhile, a lot of work is devoted to the removal of dyes and other organic pollutants using this technique. Authors need to discuss this point very carefully in a revised manuscript.
A3 Thank you for your recommendation, more information on photocatalytical degradation was added.
Photochemical oxidation is one of the most commonly used methods for the degradation of waterborne pollutants. [53] TiO2-based materials are most widely used because of their unprecedented photocatalytic activity, but other metallic and nonmetallic catalysts are giving promising results. [54–56]
Q4 All abbreviations and abbreviations must be spelled out at their first mention in the text (PXRD, TEM, etc.)
A4 Thank you for your recommendation, we added full and abbreviated terms.
Q5 I suggest that some discussion on sample synthesis should be added at the beginning of the second section of the manuscript.
A5 Thank you for your comment, we revised that part of the paper.
The present research was designed to determine whether the stainless steel (SS) composites with PbO2 and g-C3N4 could be applied for RB5 electrochemical decoloration.
Electrode materials were based on PbO2 nanoparticles, with CTAB and TBAP surfactants as templates, later combined with g-C3N4 to obtain composites tested in the removal of the selected textile dye.
Q6 Data on the method of decomposition of the dye RB5 should be added to the experimental section.
A6 Thank you for your suggestion, it looks better if we add the procedure for the degradation experiments in the third segment of this paper.
3.5. Experimental setup
Solutions of RB5 with Na2SO4 were made in concentrations ranging from 20 mg/L up to 100 mg/L. Blank probe for UV-VIS was made with 0.1 M Na2SO4. For pH adjustments, 0.1 M NaOH and 0.1 M H2SO4 were used.
The optimization of RB5 degradation experiments was held in a vessel mounted on a magnetic stirrer, with a constant starting volume of the treated solutions (60 cm3 of RB5 dissolved in Na2SO4 solution). The electrodes were immersed in the solution and connected to the voltage source PeakTech 1525. For RB5 concentration monitoring, samples were withdrawn in 10-minute intervals, up to 90 minutes of electrochemical oxidation, and analyzed by Evolution™ One/One Plus UV-Vis Spectrophotometer, Waltham, Massachusetts, USA.
The first parameter to be optimized was starting pH value (adjusted using H2SO4 and NaOH), followed by the optimization concentration of the supporting electrolyte, with a fixed pH value. When these two parameters were set right, in terms of the fastest RB5 degradation, the current density was varied to improve the degradation rate. The last criterion was the RB5 concentration and with the optimized previously stated values, the synthesized electrode materials were compared to distinguish the most excellent conditions for RB5 degradation. Data were processed using MS Excel 2016 and OriginPro 8.
Q7 The authors have to add comparative data on the efficiency of decomposition of the studied dye using other types of electrodes.
A7 Thank you for your comment. A comparison of efficiency with other described methods, as well as a tabular representation, is given in Table 2.
Table 2 provides and overview of different electrochemical systems employed to remove RB5, with focus on results with Na2SO4 as supporting electrolyte. All selected studies reported faster RB5 removal when using NaCl or KBr [20–23], but the use of these non-inert salts is discouraged in electrochemistry, because they lead to the formation of chlorinated and brominated degradation products, which are often more toxic compared to the parent compound. [22,24,25]
When excluding the results with NaCl, our study gave the fastest (under 1 h) and the highest RB5 removal (nearly 90%). The system with Ti/CoOx–RuO2–SnO2–Sb2O5 led removed only 40% of the selected dye, during 2 h of treatment in the presence of Na2SO4, but increased to over 95% when NaCl was used. [21] The next study with Ti/SnO2-Sb2O5-IrO2 was slightly improved in the same time interval without the addition of NaCl. [23] Graphite electrode lead to complete RB5 degradation and quite efficient COD removal, but this experimental setup lasted for three hours. [22]
Table 2. The RB5 degradation efficiency comparison with the literature data
|
Material |
Supporting electrolyte |
% of RB5 removal and time |
Energy consumption |
Ref |
|
RuO2/IrO2/TiO2@DSA® |
0.008 M NaCl |
~100% in 15 min |
Not reported |
[20] |
|
Ti/CoOx–RuO2–SnO2–Sb2O5 |
0.07 M Na2SO4 |
~40% in 2 h |
34.5 kWh/kg |
[21] |
|
Ti/SnO2-Sb2O5-IrO2 |
0.1 M Na2SO4 |
~60% in 2 h |
Not reported |
[23] |
|
Graphite |
0.1 M Na2SO4 |
~100% in 3 h |
Not reported |
[22] |
|
PbO2-CTAB/g-C3N4 on SS |
0.15 M Na2SO4 |
~90 % in 1 h |
0.4374 kWh/g |
This study |
a calculated as general current efficiency, based on COD removal. [26]
Q8 A graphical confirmation of the results presented should be reflected in the subsection "Stability and longevity of the method" (line 247).
A8 Thank you for this suggestion. In the revised version we provided graphical illustration of the stability studies.
Figure 8. Changes in the final absorbance during seven days of work with the same electrode.
Q9 The conclusion section should be revised and reflect the results obtained in more detail.
A9 Thank you for this suggestion. This was also commented from the reviewer 1. In the revised version we rewrote section Conclusions. This is highlighted in the revised manuscript.
In this paper, an electrochemical degradation/removal method for Reactive Black 5 azo dye was created, based on PbO2-CTAB/g-C3N4 anode material as an electrocatalyst. The parameters that were optimized include initial pH, concentration of Na2SO4 as supporting electrolyte, electric current and concentration of RB5 dye. PbO2 probes synthesized using different surfactants as templates were used separately in pristine form and as composite materials with graphitic carbon nitride. Morphological and electrochemical properties of the materials were evaluated using PXRD, TEM, SEM, CV and EIS techniques. The efficiency of the electrolytic reaction vastly depended on procedure parameters. The results showed that PbO2-CTAB/g-C3N4 exhibits the best electrocatalytic properties and the highest removal rate for RB5, with 90% of the dye removed after 60 minutes of treatment. The specific energy consumption for this process was 0.4374 kWh/g. Finally, the prepared material showed good stability and enabled the treatment of highly polluted wastewaters with excellent results, but also, due to its electrocatalytic properties, could be applied in various fields of electrochemistry, such as sensing and biosensing probes. In regards to efficiency and longevity, the obtained results suggest that the proposed method can offer a low-cost, effective and green approach in the field of environmental control, and due to its simplicity, this method has high potential for technology transfer.
Thank you for your time and effort. We appreciate all comments and conclusions. In the revised version, we have accepted all valuable suggestions and tried to improve our manuscript. All changes are highlighted in the revised version.
Reviewer 3.
This manuscript reported series of PbO2/g-C3N4 composites with different morphologies, which were synthesized by controlling various surfactants used. Electrochemical test evaluation demonstrated that the catalyst synthesized with CTAB possesses the best performance for RB5-dye and long durability. This research may help for wastewater treatment and pollutant removal in practice. Here are some comments listed below:
Q1 The introduction summarized the properties of RB-5 dye well, and listed some methods for removal of RB5. However, the summary for electrochemical oxidation was too general and only one sentence shown in Line 60-62 on page 2. The author should specify this method and give the current progress of the catalysts (i.e. advantage and disadvantage of reported catalysts and why this paper is important).
A1 Thank you for your comment. A more extensive overview of electrochemical oxidative methods is given. Also, the results of this work were compared to other methods for wastewater treatment. This can be seen in Table 2 and shows the significance of this work.
Electrochemical oxidation processes present an effective and logical choice for the development of green, time-effective and, at the same time, potent methods for the removal of various pollutants, including reactive textile dyes. [1–3] Electrocatalytic processes can be considered green primarily due to mild and environmentally acceptable working conditions (ambient pressure and temperature, work in aqueous solutions, no additional chemicals or, when they are used, they have lower toxicity). These methods may also be coupled with renewable energy sources, [4] they do not require systems for temperature or pressure control nor expensive gases and give the possibility of avoiding expensive catalysts based on precious metals, so their potential for large-scale application is enormous.
Therefore, electrochemical methods, through the rational design of the electrocatalytic setup, enable simple and practical systems for water purification and pollutant removal. Our research group proposed several approaches for the efficient removal of organic pollutants using electrooxidation methods, such as for the removal of Reactive Blue 52, [5,6] triketone herbicides [7] or ibuprofen. [8]
For electrochemical oxidative processes, anodes with high oxygen evolution potentials (non-active anodes) achieve complete mineralization of organics and are thus favored for wastewater remediation. The most commonly used non-active anode is the boron doped diamond electrode, but cheaper alternatives include various metal oxides such as TiO2, SnO2 and PbO2. [9] Metal oxides exhibit exceptional properties valuable to electrochemical processes, while featuring reduced costs, availability and environmental compatibility, and have thus found diverse applications. The properties of metal oxides can be fine-tuned by controlling their morphology, size and crystallinity, which can be achieved using template synthesis.
Lead dioxide is an inexpensive and widely available material, readily incorporated in composite materials with outstanding properties. [10] Lead oxide-based anodes gave promising performance for electrochemical degradation of organic pollutants wastewater treatment [11], as they can possess unique catalytically active surfaces, [12] good stability and a long electrode life. [13]
Graphitic carbon nitride (g-C3N4) has found numerous applications owing to its properties, such as high surface area, low cost and biocompatibility. The use of g-C3N4 in composites proved to enhance its photo- and electrocatalytic properties. Good results for the photocatalytic degradation of organic pollutants were achieved when using nanocomposites made from metal oxides and g-C3N4. [14–19] On the other hand, reports of g-C3N4 being used for water remediation in a purely electrochemical setting are scarce.
Table 2 provides and overview of different electrochemical systems employed to remove RB5, with focus on results with Na2SO4 as supporting electrolyte. All selected studies reported faster RB5 removal when using NaCl or KBr [20–23], but the use of these non-inert salts is discouraged in electrochemistry, because they lead to the formation of chlorinated and brominated degradation products, which are often more toxic compared to the parent compound. [22,24,25]
When excluding the results with NaCl, our study gave the fastest (under 1 h) and the highest RB5 removal (nearly 90%). The system with Ti/CoOx–RuO2–SnO2–Sb2O5 led removed only 40% of the selected dye, during 2 h of treatment in the presence of Na2SO4, but increased to over 95% when NaCl was used. [21] The next study with Ti/SnO2-Sb2O5-IrO2 was slightly improved in the same time interval without the addition of NaCl. [23] Graphite electrode lead to complete RB5 degradation and quite efficient COD removal, but this experimental setup lasted for three hours. [22]
Table 2. The RB5 degradation efficiency comparison with the literature data
|
Material |
Supporting electrolyte |
% of RB5 removal and time |
Energy consumption |
Ref |
|
RuO2/IrO2/TiO2@DSA® |
0.008 M NaCl |
~100% in 15 min |
Not reported |
[20] |
|
Ti/CoOx–RuO2–SnO2–Sb2O5 |
0.07 M Na2SO4 |
~40% in 2 h |
34.5 kWh/kg |
[21] |
|
Ti/SnO2-Sb2O5-IrO2 |
0.1 M Na2SO4 |
~60% in 2 h |
Not reported |
[23] |
|
Graphite |
0.1 M Na2SO4 |
~100% in 3 h |
Not reported |
[22] |
|
PbO2-CTAB/g-C3N4 on SS |
0.15 M Na2SO4 |
~90 % in 1 h |
0.4374 kWh/g |
This study |
a calculated as general current efficiency, based on COD removal. [26]
Q2 What’s the loading percentage of PbO2 on g-C3N4 and the loading amount of PbO2 on electrode? Please provide these data in the experiment section as-well. Otherwise, it makes no sense to this study if the catalysts loading percentages are not the same.
A2 Thank you for this suggestion. In original version this was omitted by mistake. In the revised version we provided detailed data about electrodes preparation.
The prepared nanomaterial suspensions, 20 µL (5 mg of PbO2/CTAB or PbO2/TBAP and 50 mg of g-C3N4 in 1 mL of dimethylformamide), were added to the electrode surface to cover its entire surface area.
Q3 Why were the CV curves were collected in Fe2+/Fe3+ solution in 0.1M KCl rather than solution without Fe2+/Fe3+?
A3 The Fe2+/3+ redox couple is commonly used in cyclic voltammetry as probe molecules, to show the electrocatalytic properties of different electrodes and new materials as modifiers of electrode surfaces. Our idea was to demonstrate the improvement of the electrocatalytic properties of the surface of a standard steel electrode by introducing new materials, as well as to compare their properties in the form of mass transfer, diffusion characteristics, resistance at the electrode/tested solution interface... These data are the easiest to obtain from testing the standard system as which is Fe2+/3+ and therefore we used this redox pair as a test probe.
Q4 In this manuscript Auxiliary electrode platinum 331 wire was used during the electrochemical evaluation. However, the dissolved Pt from wire could deposit on the counter electrode as well, which is commonly happened to ORR. So graphite electrode was typically used instead. Did the author find the difference for the electrochemical reaction if graphite electrode was adopted?
A4 We are very grateful to the reviewer for opening up the topic and a view that we have not had so far on the setup of the electrochemical experiment. No, we did not test the influence of the auxiliary electrode on the results of the electrochemical characterization of the material, but we used the usual system where we used a platinum wire as an auxiliary electrode. We found a very good study by Gregory Jerkiewicz, [57] where this phenomenon is examined in detail and in further work we will use this study when choosing an electrochemical system when recording voltammograms.
Q5 How were the pH values of electrolytes adjusted? Should provide the acid and base used exactly. Also, the author claimed the pH=2 was the best for electrochemical oxidation of RB-5 but failed to give the mechanism. Is that because of the dissolution of PdO2 in acid media?
A5 Thank you for your question. The pH was adjusted by addition of acid or base as described in the experimental setup. The optimal pH was derived by testing degradation efficiency in solutions with different starting pH values. The dissolution of lead dioxide from the electrode was not observed but may have happened only slightly, as the concentration of sulphate ions is quite high.
3.5. Experimental setup
Solutions of RB5 with Na2SO4 were made in concentrations ranging from 20 mg/L up to 100 mg/L. Blank probe for UV-VIS was made with 0.1 M Na2SO4. For pH adjustments, 0.1 M NaOH and 0.1 M H2SO4 were used.
Also, in the acidic medium, OH• radicals are produced by the anodic discharge of water in the indirect electrochemical oxidation of organic dyes at the anode. These OH• radicals adsorb onto the anode surface and oxidize the organic material. At higher pH, it could be expected that a larger amount of hydroxyl radicals would be generated, which would result in a stronger efficiency of the system. However, it was determined that the degree of ionization of the cationic dye is highly dependent on the initial pH value of the solution, which most often leads to the formation of a precipitated photochromic compound and a decrease in resistance on mass transfer at higher pH. Therefore, an acidic environment leads to better dissolution and degree of ionization. In the case of an increase in the pH value of the solution, an increased consumption of electrolytes occurs dominantly, which directly affects the conductivity of the solution. The lower pH values were not tested as recent studies showed that low pH values led to a higher dissolution rate of lead dioxide [32] and that these values are not appropriate for work with lead dioxide films, as they can cause the formation of lead sulfate. [33]
Q6 Figure 3b is blurring. Should re-draw it.
A6 Thank you for noticing, we agree with the reviewer. In the revised version we provided new Figure 3.
Figure 3. (A) RB5 removal under pH 2, 4, and 6; (B) UV-Vis spectra of RB5 solution in the range from 200 to 800 nm during 60 minutes of treatment (pH 2, 0.1 M Na2SO4, 5V) at different time intervals
Q7 Since the surfactants are hard to be removed after synthesis. Were the as-prepared electrodes with catalysts activated before collecting echem data?
The materials were synthesized following a known procedure (ref 64), which involves surfactants as templates that should be washed after synthesis. Traces of surfactants might remain in the material even after washing, but these compounds are commonly used as electrolytes in non-aqueous media and are sometimes added to electrolyte solutions in small amounts in order to improve catalytic activity, acting as phase catalysts. Therefore, their presence in solution shouldn't have any significant (and definitely not adverse) effects. Furthermore, if significant amounts of surfactants were present in the final material, they would be detected during material characterization.
References
- Jović, M.; Stanković, D.; Manojlović, D.; Anđelković, I.; Milić, A.; Dojčinović, B.; Roglić, G. Study of the Electrochemical Oxidation of Reactive Textile Dyes Using Platinum Electrode. Int J Electrochem Sci 2013, 8, 16.
- Savić, B.G.; Stanković, D.M.; Živković, S.M.; Ognjanović, M.R.; Tasić, G.S.; Mihajlović, I.J.; Brdarić, T.P. Electrochemical Oxidation of a Complex Mixture of Phenolic Compounds in the Base Media Using PbO2-GNRs Anodes. Appl. Surf. Sci. 2020, 529, 147120, doi:10.1016/j.apsusc.2020.147120.
- Žunić, M.J.; Milutinović-Nikolić, A.D.; Stanković, D.M.; Manojlović, D.D.; Jović-Jovičić, N.P.; Banković, P.T.; Mojović, Z.D.; Jovanović, D.M. Electrooxidation of P-Nitrophenol Using a Composite Organo-Smectite Clay Glassy Carbon Electrode. Appl. Surf. Sci. 2014, 313, 440–448.
- Ganiyu, S.O.; Martínez-Huitle, C.A.; Rodrigo, M.A. Renewable Energies Driven Electrochemical Wastewater/Soil Decontamination Technologies: A Critical Review of Fundamental Concepts and Applications. Appl. Catal. B Environ. 2020, 270, 118857, doi:10.1016/j.apcatb.2020.118857.
- Manojlović, D.; Lelek, K.; Roglić, G.; Zherebtsov, D.; Avdin, V.; Buskina, K.; Sakthidharan, C.; Sapozhnikov, S.; Samodurova, M.; Zakirov, R.; et al. Efficiency of Homely Synthesized Magnetite: Carbon Composite Anode toward Decolorization of Reactive Textile Dyes. Int. J. Environ. Sci. Technol. 2020, 17, 2455–2462, doi:10.1007/s13762-020-02654-8.
- Stanković, D.M.; Ognjanović, M.; Espinosa, A.; del Puerto Morales, M.; Bessais, L.; Zehani, K.; Antić, B.; Dojcinović, B. Iron Oxide Nanoflower–Based Screen Print Electrode for Enhancement Removal of Organic Dye Using Electrochemical Approach. Electrocatalysis 2019, 10, 663–671, doi:10.1007/s12678-019-00554-1.
- Jović, M.; Manojlović, D.; Stanković, D.; Dojčinović, B.; Obradović, B.; Gašić, U.; Roglić, G. Degradation of Triketone Herbicides, Mesotrione and Sulcotrione, Using Advanced Oxidation Processes. J. Hazard. Mater. 2013, 260, 1092–1099, doi:10.1016/j.jhazmat.2013.06.073.
- Marković, M.; Jović, M.; Stanković, D.; Kovačević, V.; Roglić, G.; Gojgić-Cvijović, G.; Manojlović, D. Application of Non-Thermal Plasma Reactor and Fenton Reaction for Degradation of Ibuprofen. Sci. Total Environ. 2015, 505, 1148–1155, doi:10.1016/j.scitotenv.2014.11.017.
- Jiang, Y.; Zhao, H.; Liang, J.; Yue, L.; Li, T.; Luo, Y.; Liu, Q.; Lu, S.; Asiri, A.M.; Gong, Z.; et al. Anodic Oxidation for the Degradation of Organic Pollutants: Anode Materials, Operating Conditions and Mechanisms. A Mini Review. Electrochem. Commun. 2021, 123, 106912, doi:10.1016/j.elecom.2020.106912.
- Duan, P.; Qian, C.; Wang, X.; Jia, X.; Jiao, L.; Chen, Y. Fabrication and Characterization of Ti/Polyaniline-Co/PbO2–Co for Efficient Electrochemical Degradation of Cephalexin in Secondary Effluents. Environ. Res. 2022, 214, 113842, doi:10.1016/j.envres.2022.113842.
- Xia, Y.; Wang, G.; Guo, L.; Dai, Q.; Ma, X. Electrochemical Oxidation of Acid Orange 7 Azo Dye Using a PbO2 Electrode: Parameter Optimization, Reaction Mechanism and Toxicity Evaluation. Chemosphere 2020, 241, 125010, doi:10.1016/j.chemosphere.2019.125010.
- Shmychkova, O.; Luk’yanenko, T.; Dmitrikova, L.; Velichenko, A. Modified Lead Dioxide for Organic Wastewater Treatment: Physicochemical Properties and Electrocatalytic Activity. J. Serbian Chem. Soc. 2019, 84, 187–198, doi:10.2298/JSC180712091S.
- Zhou, M.; Dai, Q.; Lei, L.; Ma, C.; Wang, D. Long Life Modified Lead Dioxide Anode for Organic Wastewater Treatment: Electrochemical Characteristics and Degradation Mechanism. Environ. Sci. Technol. 2005, 39, 363–370, doi:10.1021/es049313a.
- Su, Y.; Liu, G.; Zeng, C.; Lu, Y.; Luo, H.; Zhang, R. Carbon Quantum Dots-Decorated TiO2/g-C3N4 Film Electrode as a Photoanode with Improved Photoelectrocatalytic Performance for 1,4-Dioxane Degradation. Chemosphere 2020, 251, 126381, doi:10.1016/j.chemosphere.2020.126381.
- Kumar, A.; Kumari, A.; Sharma, G.; Du, B.; Naushad, Mu.; Stadler, F.J. Carbon Quantum Dots and Reduced Graphene Oxide Modified Self-Assembled S@C3N4/B@C3N4 Metal-Free Nano-Photocatalyst for High Performance Degradation of Chloramphenicol. J. Mol. Liq. 2020, 300, 112356, doi:10.1016/j.molliq.2019.112356.
- Hu, J.; Zhang, P.; An, W.; Liu, L.; Liang, Y.; Cui, W. In-Situ Fe-Doped g-C3N4 Heterogeneous Catalyst via Photocatalysis-Fenton Reaction with Enriched Photocatalytic Performance for Removal of Complex Wastewater. Appl. Catal. B Environ. 2019, 245, 130–142, doi:10.1016/j.apcatb.2018.12.029.
- Fang, L.; Liu, Z.; Zhou, C.; Guo, Y.; Feng, Y.; Yang, M. Degradation Mechanism of Methylene Blue by H2O2 and Synthesized Carbon Nanodots/Graphitic Carbon Nitride/Fe(II) Composite. J. Phys. Chem. C 2019, 123, 26921–26931, doi:10.1021/acs.jpcc.9b06774.
- Mohammad, A.; Ahmad, K.; Qureshi, A.; Tauqeer, Mohd.; Mobin, S.M. Zinc Oxide-Graphitic Carbon Nitride Nanohybrid as an Efficient Electrochemical Sensor and Photocatalyst. Sens. Actuators B Chem. 2018, 277, 467–476, doi:10.1016/j.snb.2018.07.086.
- Ma, S.; Xue, J.; Zhou, Y.; Zhang, Z.; Cai, Z.; Zhu, D.; Liang, S. Facile Fabrication of a Mpg-C3N4/TiO2 Heterojunction Photocatalyst with Enhanced Visible Light Photoactivity toward Organic Pollutant Degradation. RSC Adv. 2015, 5, 64976–64982, doi:10.1039/C5RA10447E.
- Jager, D.; Kupka, D.; Vaclavikova, M.; Ivanicova, L.; Gallios, G. Degradation of Reactive Black 5 by Electrochemical Oxidation. Chemosphere 2018, 190, 405–416, doi:10.1016/j.chemosphere.2017.09.126.
- Saxena, P.; Ruparelia, J. Influence of Supporting Electrolytes on Electrochemical Treatability of Reactive Black 5 Using Dimensionally Stable Anode. J. Inst. Eng. India Ser. A 2019, 100, 299–310, doi:10.1007/s40030-019-00360-4.
- Rivera, M.; Pazos, M.; Sanromán, M.Á. Development of an Electrochemical Cell for the Removal of Reactive Black 5. Desalination 2011, 274, 39–43, doi:10.1016/j.desal.2011.01.074.
- Eguiluz, K.I.B.; Hernandez-Sanchez, N.K.; Dória, A.R.; Santos, G.O.S.; Salazar-Banda, G.R.; Ponce de Leon, C. Template-Made Tailored Mesoporous Ti/SnO2-Sb2O5-IrO2 Anodes with Enhanced Activity towards Dye Removal. J. Electroanal. Chem. 2022, 910, 116153, doi:10.1016/j.jelechem.2022.116153.
- Zambrano, J.; Min, B. Comparison on Efficiency of Electrochemical Phenol Oxidation in Two Different Supporting Electrolytes (NaCl and Na2SO4) Using Pt/Ti Electrode. Environ. Technol. Innov. 2019, 15, 100382, doi:10.1016/j.eti.2019.100382.
- Carneiro, J.F.; Aquino, J.M.; Silva, A.J.; Barreiro, J.C.; Cass, Q.B.; Rocha-Filho, R.C. The Effect of the Supporting Electrolyte on the Electrooxidation of Enrofloxacin Using a Flow Cell with a BDD Anode: Kinetics and Follow-up of Oxidation Intermediates and Antimicrobial Activity. Chemosphere 2018, 206, 674–681, doi:10.1016/j.chemosphere.2018.05.031.
- Xu, L.; Guo, Z.; Du, L.; He, J. Decolourization and Degradation of C.I. Acid Red 73 by Anodic Oxidation and the Synergy Technology of Anodic Oxidation Coupling Nanofiltration. Electrochimica Acta 2013, 97, 150–159, doi:10.1016/j.electacta.2013.01.148.
- Brillas, E.; Martínez-Huitle, C.A. Decontamination of Wastewaters Containing Synthetic Organic Dyes by Electrochemical Methods. An Updated Review. Appl. Catal. B Environ. 2015, 166–167, 603–643, doi:10.1016/j.apcatb.2014.11.016.
- Yu, J.; Shu, S.; Wang, Q.; Gao, N.; Zhu, Y. Evaluation of Fe2+/Peracetic Acid to Degrade Three Typical Refractory Pollutants of Textile Wastewater. Catalysts 2022, 12, 684, doi:10.3390/catal12070684.
- Emadi, Z.; Sadeghi, R.; Forouzandeh, S.; Mohammadi-Moghadam, F.; Sadeghi, R.; Sadeghi, M. Simultaneous Anaerobic Decolorization/Degradation of Reactive Black-5 Azo Dye and Chromium(VI) Removal by Bacillus Cereus Strain MS038EH Followed by UV-C/H2O2 Post-Treatment for Detoxification of Biotransformed Products. Arch. Microbiol. 2021, 203, 4993–5009, doi:10.1007/s00203-021-02462-9.
- Saroyan, H.; Ntagiou, D.; Rekos, K.; Deliyanni, E. Reactive Black 5 Degradation on Manganese Oxides Supported on Sodium Hydroxide Modified Graphene Oxide. Appl. Sci. 2019, 9, 2167, doi:10.3390/app9102167.
- De Luca, P.; B. Nagy, J. Treatment of Water Contaminated with Reactive Black-5 Dye by Carbon Nanotubes. Materials 2020, 13, 5508, doi:10.3390/ma13235508.
- Xie, Y.; Wang, Y.; Singhal, V.; Giammar, D.E. Effects of PH and Carbonate Concentration on Dissolution Rates of the Lead Corrosion Product PbO2. Environ. Sci. Technol. 2010, 44, 1093–1099, doi:10.1021/es9026198.
- Broda, B.; Inzelt, G. Investigation of the Electrochemical Behaviour of Lead Dioxide in Aqueous Sulfuric Acid Solutions by Using the in Situ EQCM Technique. J. Solid State Electrochem. 2020, 24, 1–10, doi:10.1007/s10008-019-04450-y.
- Belal, R.M.; Zayed, M.A.; El-Sherif, R.M.; Abdel Ghany, N.A. Advanced Electrochemical Degradation of Basic Yellow 28 Textile Dye Using IrO2/Ti Meshed Electrode in Different Supporting Electrolytes. J. Electroanal. Chem. 2021, 882, 114979, doi:10.1016/j.jelechem.2021.114979.
- Sui, X.; Feng, C.; Chen, Y.; Sultana, N.; Ankeny, M.; R. Vinueza, N. Detection of Reactive Dyes from Dyed Fabrics after Soil Degradation via QuEChERS Extraction and Mass Spectrometry. Anal. Methods 2020, 12, 179–187, doi:10.1039/C9AY01603A.
- Frisch, M. ea; Trucks, G.W.; Schlegel, H.B.; Scuseria, G.E.; Robb, M.A.; Cheeseman, J.R.; Scalmani, G.; Barone, V.; Petersson, G.A.; Nakatsuji, H. Gaussian 16 2016.
- Stanton, J.F. A Chemist’s Guide to Density Functional Theory By Wolfram Koch (German Chemical Society, Frankfurt Am Main) and Max C. Holthausen (Humbolt University Berlin). Wiley-VCH: Weinheim. 2000. Xiv + 294 Pp. $79.95. ISBN 3-527-29918-1. J. Am. Chem. Soc. 2001, 123, 2701–2701, doi:10.1021/ja004799q.
- Lee, C.; Yang, W.; Parr, R.G. Development of the Colle-Salvetti Correlation-Energy Formula into a Functional of the Electron Density. Phys. Rev. B 1988, 37, 785–789, doi:10.1103/PhysRevB.37.785.
- Becke, A.D. A New Mixing of Hartree–Fock and Local Density‐functional Theories. J. Chem. Phys. 1993, 98, 1372–1377, doi:10.1063/1.464304.
- Frisch, M.J.; Pople, J.A.; Binkley, J.S. Self-Consistent Molecular Orbital Methods 25. Supplementary Functions for Gaussian Basis Sets. J. Chem. Phys. 1984, 80, 3265–3269.
- Vitaly, A.R.; RATNER, M.A.; POPLE, J.A.; REDFERN, P.C.; Larry, A.C. 6-31G* Basis Set for Third-Row Atoms. J. Chem. Phys. 2001.
- Check, C.E.; Faust, T.O.; Bailey, J.M.; Wright, B.J.; Gilbert, T.M.; Sunderlin, L.S. Addition of Polarization and Diffuse Functions to the LANL2DZ Basis Set for P-Block Elements. J. Phys. Chem. A 2001, 105, 8111–8116.
- Blaudeau, J.-P.; McGrath, M.P.; Curtiss, L.A.; Radom, L. Extension of Gaussian-2 (G2) Theory to Molecules Containing Third-Row Atoms K and Ca. J. Chem. Phys. 1997, 107, 5016–5021.
- A Geometrical Correction for the Inter- and Intra-Molecular Basis Set Superposition Error in Hartree-Fock and Density Functional Theory Calculations for Large Systems: The Journal of Chemical Physics: Vol 136, No 15 Available online: https://aip.scitation.org/doi/10.1063/1.3700154 (accessed on 30 November 2022).
- Miertuš, S.; Scrocco, E.; Tomasi, J. Electrostatic Interaction of a Solute with a Continuum. A Direct Utilizaion of AB Initio Molecular Potentials for the Prevision of Solvent Effects. Chem. Phys. 1981, 55, 117–129.
- Tomasi, J.; Mennucci, B.; Cammi, R. Quantum Mechanical Continuum Solvation Models. Chem. Rev. 2005, 105, 2999–3094.
- Barone, V.; Cossi, M. Quantum Calculation of Molecular Energies and Energy Gradients in Solution by a Conductor Solvent Model. J. Phys. Chem. A 1998, 102, 1995–2001.
- Electronic Population Analysis on LCAO‐MO Molecular Wave Functions. IV. Bonding and Antibonding in LCAO and Valence‐Bond Theories: The Journal of Chemical Physics: Vol 23, No 12 Available online: https://aip.scitation.org/doi/10.1063/1.1741877 (accessed on 30 November 2022).
- Parr, R.G.; Yang, W. Density Functional Approach to the Frontier-Electron Theory of Chemical Reactivity. J. Am. Chem. Soc. 1984, 106, 4049–4050, doi:10.1021/ja00326a036.
- Electron Density, Kohn–Sham Frontier Orbitals, and Fukui Functions: The Journal of Chemical Physics: Vol 81, No 6 Available online: https://aip.scitation.org/doi/10.1063/1.447964 (accessed on 30 November 2022).
- Perspective on “Density Functional Approach to the Frontier-Electron Theory of Chemical Reactivity” | SpringerLink Available online: https://link.springer.com/article/10.1007/s002149900093 (accessed on 30 November 2022).
- Zielinski, F.; Tognetti, V.; Joubert, L. Condensed Descriptors for Reactivity: A Methodological Study. Chem. Phys. Lett. 2012, 527, 67–72, doi:10.1016/j.cplett.2012.01.011.
- López-Ramón, M.V.; Rivera-Utrilla, J.; Sánchez-Polo, M. Photocatalytic Degradation of Organic Wastes in Water. Catalysts 2022, 12, 114, doi:10.3390/catal12020114.
- Ramírez, J.I.D.L.; Villegas, V.A.R.; Sicairos, S.P.; Guevara, E.H.; Brito Perea, M.D.C.; Sánchez, B.L. Synthesis and Characterization of Zinc Peroxide Nanoparticles for the Photodegradation of Nitrobenzene Assisted by UV-Light. Catalysts 2020, 10, 1041, doi:10.3390/catal10091041.
- Fernández-Perales, M.; Rozalen, M.; Sánchez-Polo, M.; Rivera-Utrilla, J.; López-Ramón, M.V.; Álvarez, M.A. Solar Degradation of Sulfamethazine Using RGO/Bi Composite Photocatalysts. Catalysts 2020, 10, 573, doi:10.3390/catal10050573.
- Amin Marsooli, M.; Rahimi Nasrabadi, M.; Fasihi-Ramandi, M.; Adib, K.; Pourmasoud, S.; Ahmadi, F.; Eghbali, M.; Sobhani Nasab, A.; Tomczykowa, M.; Plonska-Brzezinska, M.E. Synthesis of Magnetic Fe3O4/ZnWO4 and Fe3O4/ZnWO4/CeVO4 Nanoparticles: The Photocatalytic Effects on Organic Pollutants upon Irradiation with UV-Vis Light. Catalysts 2020, 10, 494, doi:10.3390/catal10050494.
- Jerkiewicz, G. Applicability of Platinum as a Counter-Electrode Material in Electrocatalysis Research. ACS Catal. 2022, 12, 2661–2670, doi:10.1021/acscatal.1c06040.

Reviewer 2 Report
Very interesting and well organized study. However, there are a number of comments that I hope will undoubtedly improve this manuscript: 1. Authors need to revise the list of keywords and add more specific terms directly related to this study. 2. The introduction section should be revised and more clearly demonstrate the relevance and novelty of the study. 3. In lines 54-64, the authors describe the shortcomings of the main methods for removing organic dyes. However, one of the most widely popular and used techniques - photocatalysis,was mentioned very superficially. Meanwhile, a lot of work is devoted to the removal of dyes and other organic pollutants using this technique. Authors need to discuss this point very carefully in a revised manuscript. 4. All abbreviations and abbreviations must be spelled out at their first mention in the text (PXRD, TEM, etc.) 5. I suggest that some discussion on sample synthesis should be added at the beginning of the second section of the manuscript. 6. Data on the method of decomposition of the dye RB5 should be added to the experimental section. 7. The authors have to add comparative data on the efficiency of decomposition of the studied dye using other types of electrodes. 8 A graphical confirmation of the results presented should be reflected in the subsection "Stability and longevity of the method" (line 247). 9. The conclusion section should be revised and reflect the results obtained in more detail.
Author Response

(The authors gave the same response as above.)

Reviewer 3 Report
This manuscript reported series of PbO2/g-C3N4 composites with different morphologies, which were synthesized by controlling various surfactants used. Electrochemical test evaluation demonstrated that the catalyst synthesized with CTAB possesses the best performance for RB5-dye and long durability. This research may help for wastewater treatment and pollutant removal in practice. Here are some comments listed below:
1. The introduction summarized the properties of RB-5 dye well, and listed some methods for removal of RB5. However, the summary for electrochemical oxidation was too general and only one sentence shown in Line 60-62 on page 2. The author should specify this method and give the current progress of the catalysts (i.e. advantage and disadvantage of reported catalysts and why this paper is important).
2. What’s the loading percentage of PbO2 on g-C3N4 and the loading amount of PbO2 on electrode? Please provide these data in the experiment section as-well. Otherwise, it makes no sense to this study if the catalysts loading percentages are not the same.
3. Why were the CV curves were collected in Fe2+/Fe3+ solution in 0.1M KCl rather than solution without Fe2+/Fe3+?
4. In this manuscript Auxiliary electrode platinum 331 wire was used during the electrochemical evaluation. However, the dissolved Pt from wire could deposit on the counter electrode as well, which is commonly happened to ORR. So graphite electrode was typically used instead. Did the author find the difference for the electrochemical reaction if graphite electrode was adopted?
5. How were the pH values of electrolytes adjusted? Should provide the acid and base used exactly. Also, the author claimed the pH=2 was the best for electrochemical oxidation of RB-5 but failed to give the mechanism. Is that because of the dissolution of PdO2 in acid media??
6. Figure 3b is blurring. Should re-draw it.
7. Since the surfactants are hard to be removed after synthesis. Were the as-prepared electrodes with catalysts activated before collecting echem data?
Author Response

(The authors gave the same response as above.)

Round 2
Reviewer 1 Report
Authors have made significant improvement to the manuscript. In present form, it may be accepted for publication.
Reviewer 2 Report
Authors responds for all my querries and thr final version of revised manuscript could be recommended for publication